# Pre-activated nanoparticles with persistent luminescence for deep tumor photodynamic therapy in gallbladder cancer

Sarun Juengpanich[1,2,3,11], Shijie Li[1,2,11], Taorui Yang[4,11], Tianao Xie[1,2], Jiadong Chen[5], Yukai Shan[1], Jiyoung Lee[6], Ziyi Lu[1], Tianen Chen[1], Bin Zhang[1], Jiasheng Cao[1], Jiahao Hu[1], Jicheng Yu[1,6,7,8,9], Yanfang Wang[6], Win Topatana ®[1,2] ✉, Zhen Gu ®[1,6,7,8,9,10] ✉, Xiujun Cai ®[1,2,3] ✉ & Mingyu Chen ®[1,2,3] ✉

Phototherapy of deep tumors still suffers from many obstacles, such as limited near-infrared (NIR) tissue penetration depth and low accumulation efficiency within the target sites. Herein, stimuli-sensitive tumor-targeted photodynamic nanoparticles (STPNs) with persistent luminescence for the treatment of deep tumors are reported. Purpurin 18 (Pu18), a porphyrin derivative, is utilized as a photosensitizer to produce persistent luminescence in STPNs, while lanthanide-doped upconversion nanoparticles (UCNPs) exhibit bioimaging properties and possess high photostability that can enhance photosensitizer efficacy. STPNs are initially stimulated by NIR irradiation before intravenous administration and accumulate at the tumor site to enter the cells through the HER2 receptor. Due to Pu18 afterglow luminescence properties, STPNs can continuously generate ROS to inhibit NFκB nuclear translocation, leading to tumor cell apoptosis. Moreover, STPNs can be used for diagnostic purposes through MRI and intraoperative NIR navigation. STPNs exceptional antitumor properties combined the advantages of UCNPs and persistent luminescence, representing a promising phototherapeutic strategy for deep tumors.

Gallbladder cancer (GBC) is a deep-seated malignancy that presents with vague symptoms in its early stages. Most GBC patients are diagnosed at an advanced stage, with a 5-year survival rate of <5%[1–3]. Although surgical resection is one of the most effective treatments for GBC, over 60% percent of patients have recurrence within 5 years[4]. Conventional chemotherapy and radiotherapy are unable to prolong the survival of patients with recurrent or advanced GBC. Thus, it is imperative to develop an effective diagnostic and therapeutic strategy for GBC.

Photodynamic therapy (PDT) has attracted increasing interest as a noninvasive therapeutic option for cancer due to its remarkable therapeutic efficacy[5,6]. It is frequently used to treat superficial diseases

[1]Department of General Surgery, Sir Run-Run Shaw Hospital, Zhejiang University, 310016 Hangzhou, China. [2]School of Medicine, Zhejiang University, 310058 Hangzhou, China. [3]National Engineering Research Center of Innovation and Application of Minimally Invasive Instruments, Sir Run-Run Shaw Hospital, Zhejiang University, 310016 Hangzhou, China. [4]Department of Chemistry, Zhejiang Sci-Tech University, 310018 Hangzhou, China. [5]Department of Chemistry, Zhejiang University, 310016 Hangzhou, China. [6]Zhejiang Provincial Key Laboratory for Advanced Drug Delivery Systems, College of Pharmaceutical Sciences, Zhejiang University, 310058 Hangzhou, China. [7]National Key Laboratory of Advanced Drug Delivery and Release Systems, Zhejiang University, 310058 Hangzhou, China. [8]Liangzhu Laboratory, Zhejiang University Medical Center, 311121 Hangzhou, China. [9]Jinhua Institute of Zhejiang University, 321299 Jinhua, China. [10]MOE Key Laboratory of Macromolecular Synthesis and Functionalization, Department of Polymer Science and Engineering, Zhejiang University, 310027 Hangzhou, China. [11]These authors contributed equally: Sarun Juengpanich, Shijie Li, Taorui Yang. ✉e-mail: win.topatana@zju.edu.cn; guzhen@zju.edu.cn; srrsh_cxj@zju.edu.cn; mychen@zju.edu.cn

or cancers, such as actinic keratosis, basal cell skin cancer, and squamous cell skin cancer[7,8]. However, the antitumor therapeutic efficacy for deep tumors is insufficient due to the limited tissue penetration depth of NIR, which cannot reach the tumor site to activate the phototherapeutic agents[9,10]. Although several studies have been reported to improve the penetration depth by expanding the wavelengths from NIR-I to NIR-II[11,12], the treatment of deep tumors such as GBC requires an impenetrable depth of over 4 cm, which remains challenging for NIR-mediated therapy.

Persistent luminescence is a light-excitation-free modality induced by the gradual release of photons from energy traps within the materials[13], which can be a promising non-invasive antitumor phototherapeutic approach beyond the aforementioned limitations of NIR penetration. Several organic composites, inorganic metal compounds, metal-organic frameworks, and polymers have been reported to possess persistent luminescence[14–16]. Porphyrin derivatives, such as Purpurin 18 (Pu18), were often utilized as photosensitizers in prior research, while recent studies indicated that porphyrin derivatives also exhibit inherent persistent luminescence after excitation light ceases[17,18]. Upon excitation, Pu18 transfers the absorbed energy to oxygen for singlet oxygen ($^1O_2$) generation, which then oxidizes the vinylene bond (C=C) to create a Pu18-dioxetane intermediate[19]. Spontaneous decomposition of the Pu18-dioxetane can transfer sufficient energy to excite Pu18, which is further returned to its ground state by the emission of afterglow luminescence. Interestingly, upon the incorporation of lanthanide ion-doped up-conversion nanoparticles (UCNPs) with Pu18, we found a sustained NIR-emitting luminescence after the cessation of 980 nm laser irradiation. This combined the benefits of both UCNPs and persistent luminescence, providing a novel strategy for biomedical applications in bioimaging (MRI, CT, etc.) and theranostics[20–22].

In this work, we develop stimuli-sensitive tumor-targeted photodynamic nanoparticles (STPNs) consisting of UCNPs, stimuli-responsive polymeric ligands (SPLs), and trastuzumab (TZB), for HER2-targeted GBC therapy (Fig. 1a). STPNs can aggregate on GBC cell membranes and internalize via the HER2 receptor. Due to the secondary excitation energy generated by the UCNPs and the persistent luminescence of porphyrin derivatives[17,18], STPNs can be excited by 980 nm laser irradiation prior to intravenous administration. At physiological conditions (pH 7.4), the photosensitizers aggregated within STPNs are self-quenched and lack detectable photoactivity. Upon entering the acidic tumor microenvironment (TME), STPNs disassemble into isolated UCNPs, and then the upconverted emission light from UCNPs can induce the photoactivity of Pu18 to generate reactive oxygen species (ROS) for antitumor therapy (Fig. 1b). In addition, STPNs have the potential as contrast agents for MRI due to the paramagnetic gadolinium (Gd) doped within the UCNPs.

## Results

### Synthesis and characterization of STPNs

We first prepared Pu18-grafted carboxyl-poly(ethylene glycol)-block-poly(L-glutamic acid) (COOH-PEG-PLG-Pu18) as a template, in which the flanking carboxyl groups can react with primary amines through amidation (Supplementary Fig. 1). The successful conjugation of COOH-PEG-PLG-Pu18 was confirmed by $^1$H-NMR (Supplementary Figs. 2–5). Next, we further engineered SPLs by conjugating COOH-PEG-PLG-Pu18 with an ionizable group, 1-(3-aminopropyl) imidazole (API), to induce pH sensitivity to the acidic TME, and adding 3-phenyl-1-propylamine (PPA) as high-affinity anchors for the UCNPs to facilitate self-assembly. The successful conjugation of API and PPA to the platform ligand was confirmed via $^1$H-NMR (Supplementary Fig. 6). Subsequently, the hydrophobicity of Pluronic F127 (F127) was added to SPL-modified UCNPs to provide colloidal stability[23]. Finally, STPNs were formed by amidation reaction between the amine group in TZB with the terminal carboxyl group of the SPLs and were verified via FTIR

spectrometry in Supplementary Fig. 7. Thus, STPNs can enter GBC cells via TZB, a monoclonal antibody that acts as a targeting ligand by specifically binding to the HER2 receptor on the surface of the tumor cells. Meanwhile, due to the secondary excitation energy generated by the UCNPs and the persistent luminescence of Pu18, STPNs can be pre-activated with 980 nm laser irradiation before systemic administration. Further descriptions of the synthetic pathways and characterizations are provided in the supplementary information (Supplementary Figs. 8–10). The stimuli-insensitive tumor-targeted photodynamic nanoparticles (SITPNs) and stimuli-sensitive photodynamic nanoparticles (SPNs) were synthesized utilizing the same synthetic procedure but without imidazole-containing ligands or TZB conjugation, respectively.

NaGdF$_4$:Yb,Tm@CaF$_2$:Eu core@shell UCNPs were used as nanotransducers for PDT and the activation of Pu18. The UCNPs were synthesized in a mixed organic solution containing oleic acid and 1-octadecene, as described in previous work[24]. As shown by transmission electron microscopy (TEM), energy dispersive spectrometry (EDS) mapping analysis, high-angle annular dark-field scanning transmission electron microscopy (HAADF-STEM), and dynamic light scattering (DLS) (Fig. 2a and Supplementary Figs. 11 and 12), NaGdF$_4$:Yb,Tm@CaF$_2$:Eu UCNPs have a square shape with uniform size and excellent dispersion. All elements are uniformly distributed in the UCNPs, which can be deduced from the overlapping of these elements. The presence of Ca and Eu elements indicates the successful synthesis of the core@shell structure of the UCNPs. As depicted in Supplementary Fig. 13, the thickness of the CaF$_2$ shell is 2.95 ± 0.53 nm. Furthermore, X-ray diffraction (XRD) analysis was performed to reveal the crystalline structure of the synthesized UCNPs (Supplementary Fig. 14). The XRD patterns of NaGdF$_4$:Yb,Tm core nanoparticles show the typical diffraction peaks of (Na,Gd,Yb,Tm)F$_2$ (JCPDS No. 48–1850) with a hexagonal phase crystal structure, in which the peaks at 16.8°, 29.6°, and 30.4° correspond to the (100), (110), and (101) lattice planes, respectively. In contrast, on the XRD peaks of NaGdF$_4$:Yb,Tm@CaF$_2$:Eu, a new (Ca,Eu)F$_2$ phase distinct from (Na,Gd,Yb,Tm)F$_2$ was observed, demonstrating that the core-shell structure has been obtained.

Moreover, the shell formation on the UCNP core can dramatically improve UCL intensity[25,26]. The UCL spectra for NaGdF$_4$:Yb,Tm@CaF$_2$:Eu core@shell UCNPs under 980 nm NIR irradiation is significantly higher than NaGdF$_4$:Yb,Tm core nanoparticles. As demonstrated in Supplementary Fig. 15, the minimal power of luminescent light achievable with the UCNPs is approximately 0.3 W cm$^{-2}$. After 980 nm laser irradiation that corresponds to the absorption of Yb$^{3+}$, the energy transfer from excited Yb$^{3+}$ ions to Tm$^{3+}$ ions in the NaGdF$_4$:Yb,Tm core leads to intense blue emissions at 453 nm and 477 nm due to the multiple-photon upconversion of Tm$^{3+}$ ions. The excited energy is then transferred from the $^1I_6$ level of Tm$^{3+}$ ions to the $^6P_{7/2}$ level of Gd$^{3+}$ ions, before transferring to the $^5D_0$ level of Eu$^{3+}$ ions. Subsequently, Eu$^{3+}$ ions will give out sharp emission peaks at 593 nm, 618 nm, and 700 nm, which corresponds to $^5D_0{\rightarrow}^7F_1$, $^5D_0{\rightarrow}^7F_2$, and $^5D_0{\rightarrow}^7F_4$ transitions of Eu$^{3+}$ ions, respectively (Supplementary Fig. 16). Thus, the NaGdF$_4$:Yb,Tm@CaF$_2$:Eu core@shell UCNPs have a reddish purple light emission due to the blue emission peaks of Tm$^{3+}$ ions that co-exist with the Eu$^{3+}$ ions emission peaks. Additionally, Gd-based UCNPs have great potential as contrast agents for T$_1$-weighted MRI[27,28]. As shown in Supplementary Fig. 17, T$_1$-weighted MR images of NaGdF$_4$:Yb,Tm@CaF$_2$:Eu UCNPs and STPNs became brighter as Gd concentration increased.

As depicted in the TEM images (Fig. 2a and Supplementary Fig. 18), multiple UCNPs combine to form STPNs at pH 7.4 and disintegrate to form isolated UCNPs under acidic conditions. At the physiological blood pH of 7.4, STPNs are negative-charged with a size of ~150 nm and lack detectable photoactivity (Fig. 2b, c). However, their surface charge can rapidly change from negative to positive upon exposure to the acidic TME. This is due to the decrease in pH inside the

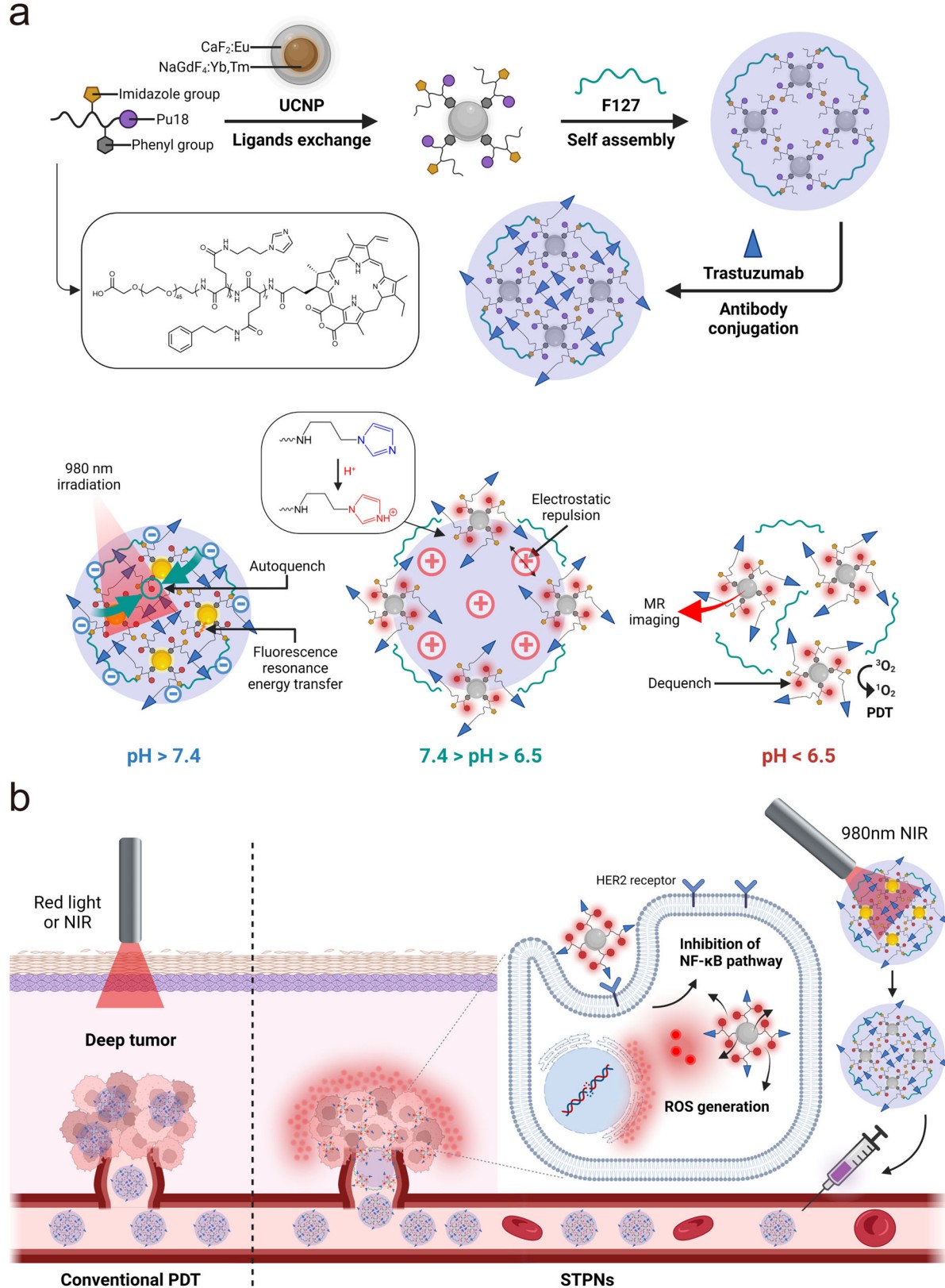

**Fig. 1 | Design and mechanism of STPNs. a** Schematic representation of self-assembled STPNs and pH-responsive mechanism of STPNs. **b** Schematic illustration of STPNs activation for deep tumor phototherapy.

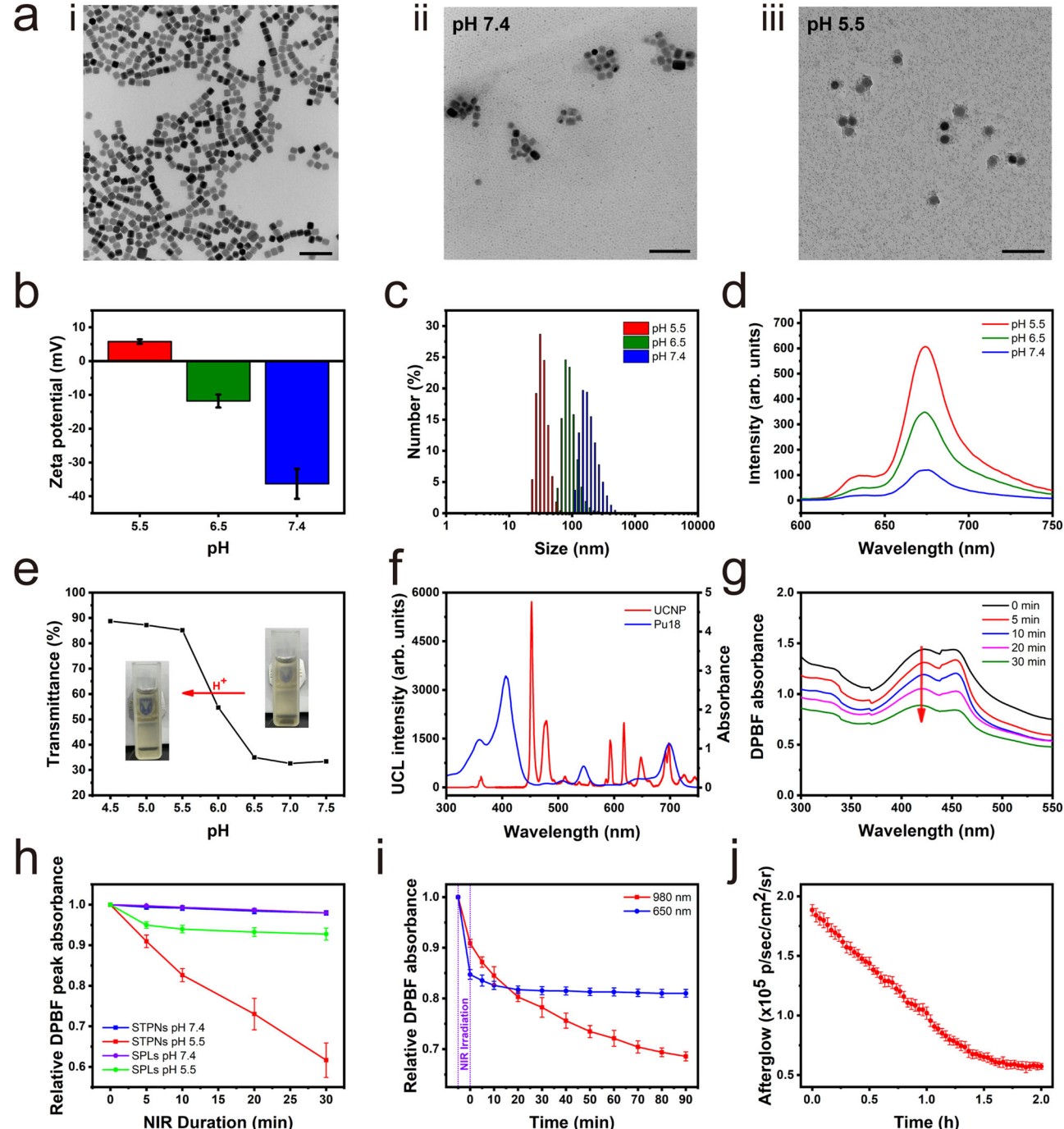

**Fig. 2 | Characterization of STPNs. a** TEM image of UCNP (i), STPNs at pH 7.4 (ii), and STPNs at pH 5.5 (iii) (scale bar = 100 nm). **b** STPNs zeta potential changes at different pH. **c** STPNs particle size distribution at different pH. **d** Pu18 fluorescence intensity from STPNs at different pH ($\lambda_{ex}$ = 400 nm, $\lambda_{em}$ = 675 nm). **e** STPNs pH-dependent transmittance changes; Inset: STPNs photographs at different pH. **f** Pu18 absorbance spectrum and UCNP UCL spectrum. **g** STPNs DPBF absorbance spectrum changes under 980 nm laser irradiation at pH 5.5. **h** pH-dependent DPBF absorbance of STPNs and SPLs under 980 nm laser irradiation at different pH. **i** STPNs (pH 5.5) $^1O_2$ generation after 5 min of 980 or 650 nm laser irradiation (2.0 W cm$^{-2}$). **j** Afterglow luminescence decay of STPNs (pH 5.5) at room temperature after 5 min of 980 nm laser irradiation (2.0 W cm$^{-2}$). The data are represented as mean ± SD (n = 3 independent experiments).

TME that leads to an increase in imidazole ionization, resulting in charge reversal and swelling, thus increasing the electrostatic repulsive force between UCNPs inside STPNs. STPNs start to swell and partially dissociate at the extracellular tumor pH of 6.5, which are further disintegrated into isolated UCNPs at the intracellular tumor pH of 5.5, as the hydrophobic interactions within STPNs became lower than the repulsion force between ionized unimers. This disassembly mechanism facilitates the dissociation of the self-quenched

aggregated photosensitizer into dequenched free molecules[29], which substantially enhances the photoactivity of Pu18 (Fig. 2d). Upon NIR irradiation, the upconverted emission light from the UCNPs will stimulate the photoactivity of free Pu18 within the acidic TME. In addition, the intense UCL of STPNs may be employed for image-guided PDT. We further investigated the disassembly mechanism by observing spectral changes. As pH decreases, STPNs absorbance decreases gradually owing to an increase in transmittance (Fig. 2e and

Supplementary Fig. 19). Notably, changes in pH do not affect the size and Pu18 fluorescence intensity of SITPNs (Supplementary Fig. 20). These findings indicate the pH-dependent characteristic of STPNs.

Next, we examined whether the UCL from the UCNPs can activate free Pu18 under the acidic TME. As depicted in Fig. 2f, the red emission from the UCNPs coincides with one of the absorbance peaks of Pu18, demonstrating that UCNPs may serve as transducers for the activation of Pu18. Furthermore, we investigated the pH-dependent ROS generation induced by Pu18 upon NIR irradiation using 1,3-diphenyliso-benzofuran (DPBF) as the $^1O_2$ indicator, in which $^1O_2$ irreversibly quenches its absorption[30,31]. Under acidic pH conditions, the DPBF absorption intensity gradually decreases with NIR irradiation duration (Fig. 2g, h), indicating that Pu18 efficiently produced $^1O_2$. On the contrary, the DPBF absorption minimally decreased under NIR irradiation at neutral pH 7.4 containing STPNs or SPLs (Supplementary Fig. 21). These findings describe the on-off activity of pH-activated PDT in STPNs when the pH is frequently changed between 7.4 and 5.5 (Supplementary Fig. 22). Moreover, DPBF was also employed to determine $^1O_2$ generation after the cessation of NIR laser irradiation. As shown in Fig. 2i, STPNs can sustainably produce $^1O_2$ for over 90 min after the cessation of 980 nm laser irradiation. However, under 650 nm laser irradiation, $^1O_2$ was rapidly generated and then gradually terminated within 20 min. In contrast, the excitation of STPNs under 980 nm laser irradiation will initially stimulate UCNPs, which will then generate secondary excitation energy to activate Pu18 for $^1O_2$ production. The $^1O_2$ quantum yield of STPNs under continuous 980 nm laser irradiation is 8%, while the $^1O_2$ quantum yield 30 min after stopping 980 nm laser irradiation is 1% (Supplementary Fig. 23). Similarly, the afterglow of STPNs has a half-life of over 1 h (Fig. 2j and Supplementary Fig. 24), which is an order of magnitude longer than the afterglow of organic semiconducting polymeric afterglow systems[14], indicating that the active structure formed by Pu18 decomposes at a slower pace. In addition, electron spin resonance (ESR) spectra with 5,5-dimethyl-1-pyrroline-N-oxide (DMPO) spin traps were monitored to capture the short-lived $^1O_2$[32], in which a distinct signal can be observed after 980 nm irradiation, verifying the production of free radicals (Supplementary Fig. 25).

## STPNs GBC cellular uptake and cytotoxicity in vitro

Compared to other biliary tract cancer cell lines, GBC-SD and EH-GB1 showed significant HER2 expression and were selected as the two cell lines for subsequent relevant research (Supplementary Fig. 26). As depicted by fluorescence microscopy imaging, STPNs entered the cells in large quantities and aggregated around the nuclei of both GBC cell lines (GBC-SD and EH-GB1) under the acidic TME at 1 h (Fig. 3a). Due to the recovery of Pu18 photoactivity resulting from the dissociation of STPNs under acidic pH, the fluorescence intensity at pH 6.5 is higher than at pH 7.4. Therefore, the presence of Pu18 signals from STPNs surrounding DAPI-stained cells in fluorescence microscopy images showed greater cellular uptake at pH 6.5 after 1 h. Notably, due to the degradation of STPNs, the fluorescence intensity of Pu18 was approximately identical in acidic and neutral environments after 4 h. By observing STPNs using TEM, we discovered that STPNs are rapidly endocytosed into the cell. Once STPNs enter the acidic TME, it disintegrates into isolated UCNPs and accumulates around the cell nucleus, consistent with the results of fluorescence microscopy (Fig. 3b). This indicates that TZB on STPNs can selectively bind with the HER2 receptor on the cell membrane of GBC-SD and EH-GB1 cells, thereby promoting the endocytosis of STPNs. Moreover, the cytotoxicity of STPNs and the encapsulated Pu18 under NIR irradiation was examined. STPNs exhibited insignificant cytotoxicity compared to TZB mono-therapy at the same dose in both GBC cell lines (Fig. 3c). In addition, cytotoxicity induced by different concentrations of Pu18 with or without NIR was also identified (Fig. 3d).

## In vitro antitumor mechanism and activity of STPNs

STPNs can provide an enhanced antitumor ability due to the synergistic effect. ROS generated by STPNs after 980 nm laser irradiation could induce apoptosis in tumor cells through multiple signaling pathways, including NFκB[33,34]. In addition, TZB on STPNs can bind with HER2 to inhibit NFκB nuclear translocation[35,36], thereby enhancing tumor cell apoptosis (Fig. 4a). The specific treatment mechanism of STPNs was further investigated and verified in GBC cell lines (GBC-SD and EH-GB1). As depicted in Fig. 4b, cytoplasmic p65 and p50 expression increased after STPNs+NIR treatment, whereas nuclear p65 and p50 expression decreased, demonstrating a more significant suppression of NFκB signaling compared to the SPNs+NIR or STPNs group. Furthermore, NIR irradiation induced ~2–4-fold of ROS in Pu18, resulting in damage to both the associated proteins and the whole tumor cells (Fig. 4c and Supplementary Fig. 27). Overall, the antitumor activity of STPNs is primarily due to the generated ROS, with the inhibition of NFκB acting as a synergistic partner to enhance the tumor-killing effect.

Using a Cell Counting Kit-8 (CCK-8) assay, proliferation assessments in GBC cell lines (GBC-SD and EH-GB1) were performed to further investigate the synergistic effects and antitumor activity of STPNs +NIR. The half-maximal inhibitory concentration (IC50) of STPNs+NIR against GBC-SD cells is 27.6 nM (in terms of TZB dose), an 8.5-fold decrease, compared with 234.2 nM for TZB alone (Supplementary Fig. 28). Similar results were obtained with EH-GB1. Next, TZB concentrations below the IC50 (50 nM) were chosen to verify the antitumor ability of STPNs (Supplementary Fig. 29). Equal concentrations of STPNs and free TZB exhibited comparable cytotoxicity, indicating the excellent biocompatibility of STPNs. A significant anti-proliferative response was detected in the STPNs group upon NIR irradiation, whereas drug-free SPNs irradiated with NIR did not exhibit obvious suppression of proliferation, demonstrating the efficacy of TZB-PDT (Fig. 4d). The 72 h growth curve of GBC-SD or EH-GB1 also demonstrated that the STPNs+NIR group had the most significant anti-proliferative effects (Supplementary Fig. 30). Furthermore, flow cytometry measurements of apoptosis conditions were comparable to the anti-proliferation assessments (Supplementary Fig. 31). Thus, the synergistic effects of TZB and NIR irradiation were only observed in STPNs. In addition, tumor cell colony formation was significantly inhibited in cells treated with STPNs+NIR, further confirming the anti-proliferative effects (Supplementary Fig. 32).

## In vivo anti-orthotopic GBC tumor activity of STPNs

To further evaluate the antitumor effect of NIR-activatable STPNs on an orthotopic GBC animal model, we administered different formulations, including PBS, TZB, STPNs, and STPNs with NIR (irradiated with a 980 nm laser source for 5 min at a power density of 2.0 W cm$^{-2}$ before injection), to orthotopic GBC-SD-luc bearing mice by tail vein injection with the same TZB concentrations (2.5 mg kg$^{-1}$) and monitored the tumors by bioluminescence (BL) on days 10, 17, 24, 31, and 38 after the tumor implantation (Fig. 5a). As shown in Fig. 5b, the tumor BL intensity in the livers of mice treated with STPNs+NIR gradually decreased and disappeared, demonstrating the excellent tumor suppressive efficacy of this treatment. In contrast, mice receiving TZB monotherapy or STPNs exhibited a less effective antitumor effect, while PBS treatment minimally inhibited tumor proliferation. Notably, due to the weak tissue penetration of the bioluminescent agent, in vivo imaging can only show moderate tumor fluorescence intensity and cannot fully reflect the true progression of the tumor. For instance, unlike the bioluminescent images, the representative image of livers excised from mice treated with STPNs+NIR showed the presence of fewer, more dispersed, small tumors on the liver (Supplementary Fig. 33). Similarly, both the TZB and STPNs groups had more tumor metastases on the liver. In general, the two types of images show the

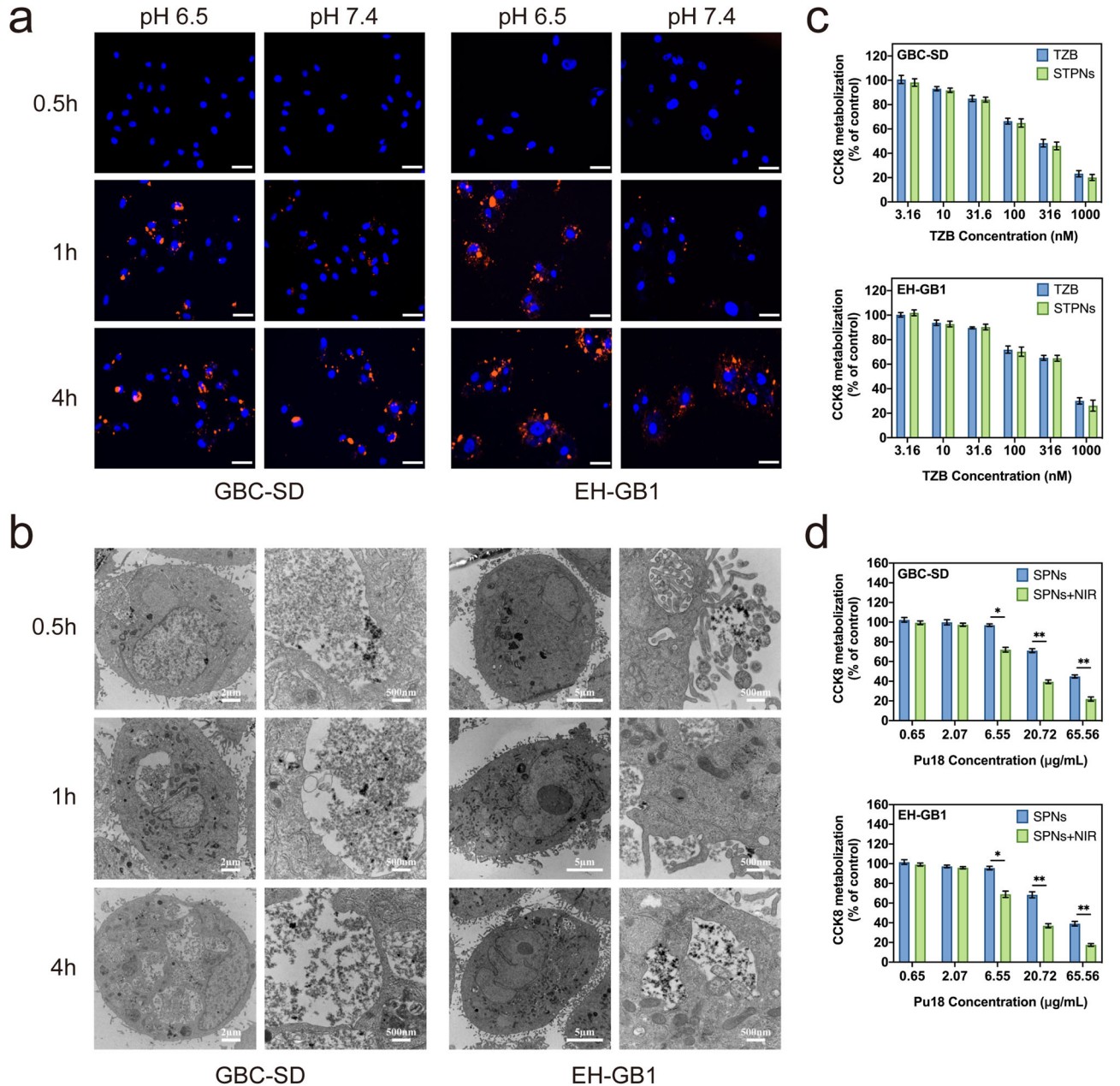

**Fig. 3 | In vivo interactions of STPNs with GBC cells. a** Fluorescence images of STPNs cellular uptake by GBC-SD or EH-GB1 cells under physiological pH 7.4 and acidic tumor microenvironment pH 6.5 at different time points. DAPI-stained cells are depicted in blue; STPNs fluorescence signals are depicted in red (scale bar = 100 μm). **b** Cryo-TEM images of STPNs cellular uptake by GBC-SD or EH-GB1 cells at different time points. **c** 48 h CCK-8 assays of GBC-SD or EH-GB1 cells treated with different concentrations of TZB (equivalent to dosage in STPNs) or STPNs. **d** 48 h CCK-8 assays of GBC-SD or EH-GB1 cells exposed to different concentrations of SPNs or SPNs with 980 nm laser irradiation (2.0 W cm⁻², 5 min with every 1 min interval). The data are represented as mean ± SD ($n = 3$ independent experiments, Student's $t$ test). *$P < 0.05$, **$P < 0.01$.

same growth pattern of orthotopic tumors, allowing us to accurately observe the antitumor effect of STPNs in vivo.

Furthermore, no discernible weight loss was observed in mice treated with NIR-activated STPNs, indicating that the intervention had minimal adverse effects (Fig. 5c). In contrast, mice treated with PBS had a significant loss in body weight, which was mainly due to tumor-induced liver dysfunction. Notably, the median survival duration of mice treated with NIR-activated STPNs was 76 days, significantly longer than that of mice treated with PBS (42 days), TZB (48 days), or inactivated STPNs (50 days) (Fig. 5d). Hematoxylin-eosin (H&E) staining of the whole liver revealed that NIR-activated STPNs significantly reduced the number and size of tumors in the liver, which was consistent with

the smallest tumor BL intensity results. Similarly, Ki-67 immunolabeling exhibited an apparent suppression of cancer cell growth in the same group, confirming the superior antitumor efficacy of STPNs +NIR (Fig. 5e).

## STPNs tumor-targeting ability and biosafety

To evaluate the tumor-targeting ability of STPNs, free Pu18, SPNs, and STPNs were injected intravenously for in vivo fluorescence (FL) imaging of nanoparticle biodistribution. At 1 h post-injection, both SPNs and STPNs exhibited FL accumulation at the tumor site compared to free Pu18, but the FL intensity in STPNs was significantly greater than that of SPNs, indicating the robust tumor-targeting and retention

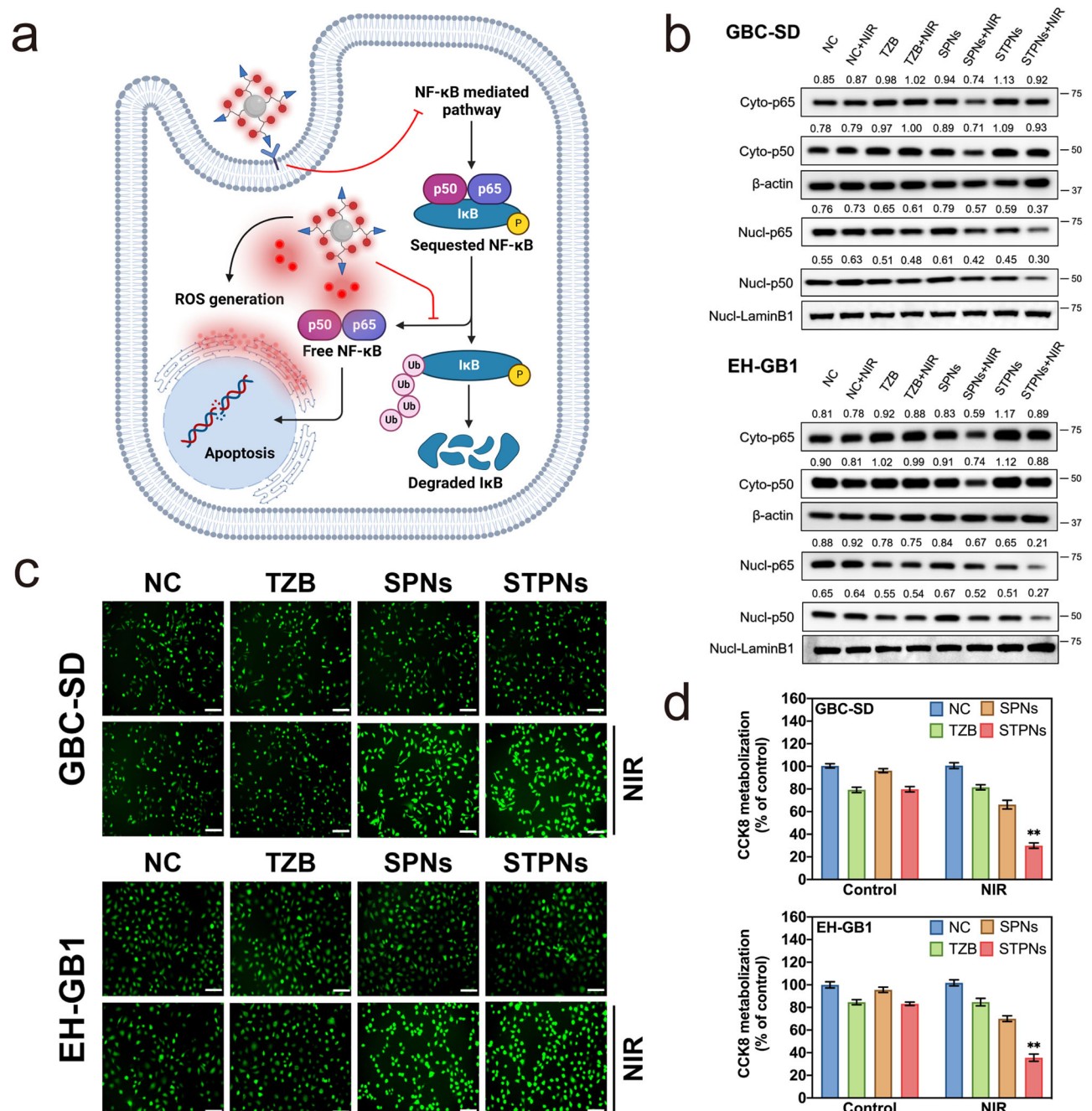

**Fig. 4 | Antitumor activity of STPNs in vitro. a** Schematic illustration of STPNs underlying molecular mechanism. **b** Western blot for NFκB p65, NFκB p50, β-actin, and Lamin B1 in the cytoplasm or/and nucleus of GBC-SD or EH-GB1 cells exposed to normal media, TZB, SPNs, or STPNs with/without 980 nm laser irradiation (2.0 W cm⁻², 5 min with every 1 min interval). **c** Intracellular ROS observation of GBC-SD or EH-GB1 cells treated as indicated (scale bar = 100 μm). **d** 48 h CCK-8 assays of GBC-SD or EH-GB1 cells exposed to normal media, TZB, SPNs, or STPNs with/without 980 nm laser irradiation (2.0 W cm⁻², 5 min with every 1 min interval). The data are represented as mean ± SD (*n* = 3 independent experiments, one-way ANOVA and Tukey's multiple comparison test). \*\**P* < 0.01.

capabilities of STPNs via HER2-mediated endocytosis (Fig. 6a). The livers with their corresponding tumors and other organs from nude mice (heart, lung, liver, kidney, and spleen) were also excised at 0.5, 1, 2, 4, 12, 24 h post-injection of STPNs for ex vivo imaging (Supplementary Fig. 34). As the primary metabolic organ, the FL intensity at different time points revealed that STPNs are predominantly distributed in the liver. STPNs were completely degraded after 24 h, whereas the majority of nanoparticles were degraded at a slower rate of over 48 h after administration[37]. At 1 h post-injection, more than half of the Pu18 afterglow in STPNs remained and can continue to generate ROS in tumor cells (Fig. 2j). Furthermore, the STPNs+NIR group

showed significant antitumor effects in the animal experiments described above, demonstrating the ability of STPNs to be pre-activated prior to systemic administration. Next, we also utilized the same NIR imaging system to simulate the FL imaging of the surgical navigation system 1 h after STPNs administration. As demonstrated in Fig. 6b, FL intensity accurately reflected the tumor site, confirming its superior positional accuracy. Similarly, MR imaging of STPNs has acquired the same tumor localization capability. For biosafety assessment, we evaluated the systemic toxicity of STPNs+NIR on day 40. Healthy mice exhibited no change in hematological indicators or liver function compared to the control group (Fig. 6c, d). H&E staining

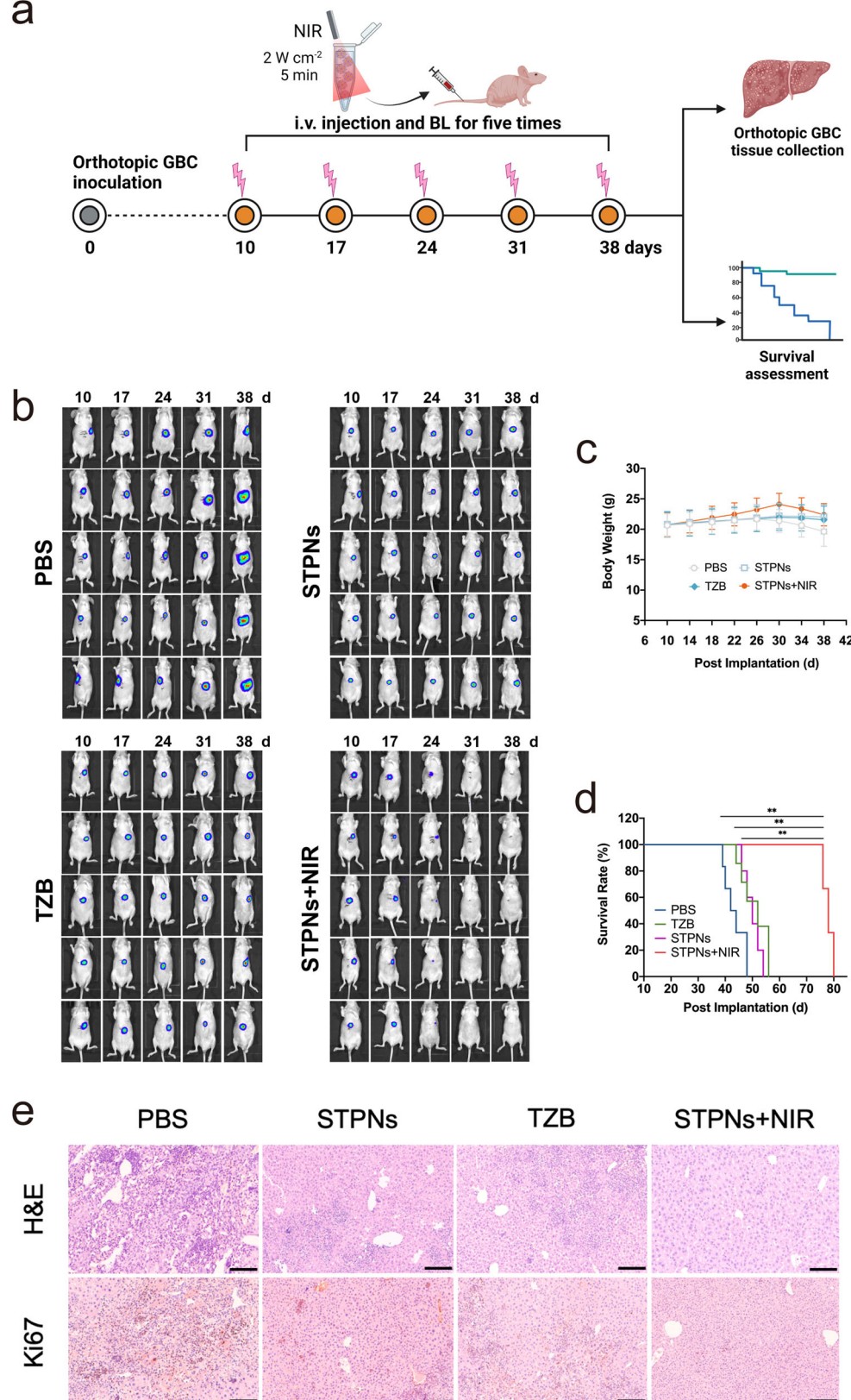

**Fig. 5 | Antitumor therapeutic effects of STPNs in vivo. a** Schematic illustration of the establishment and treatment of orthotopic GBC-bearing mice. **b** In vivo bioluminescence (BL) images of orthotopic GBC in live mice receiving PBS, TZB, STPNs, or STPNs irradiated by 980 nm laser source before injection at a power density of 2.0 W cm$^{-2}$ for 5 min. **c** Body weight changes of orthotopic GBC model mice in different groups (*n* = 5 biologically independent samples). Data are presented as mean ± SD. **d** Kaplan–Meier analysis of orthotopic GBC model mice in different groups (*n* = 5 biologically independent samples, one-way ANOVA and Tukey's multiple comparison test). **P < 0.01. **e** Representative images of H&E and Ki67 staining of orthotopic GBC tissues in different groups after treatments (scale bar = 250 μm).

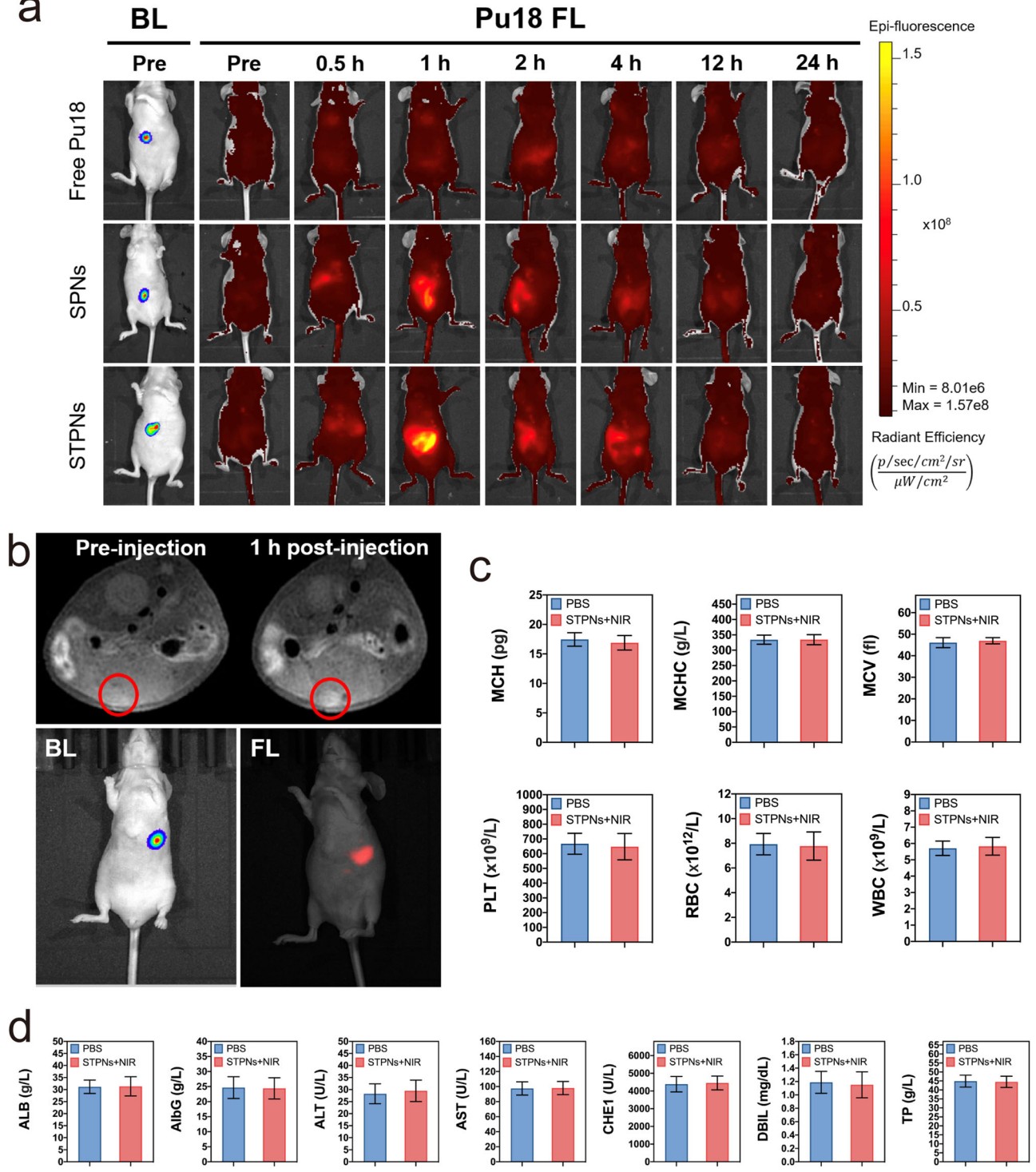

**Fig. 6 | STPNs tumor-targeting ability, bioimaging, and biosafety. a** IVIS images of tumor-bearing mice depicting tumor bioluminescence (BL) and Pu18 fluorescence (FL) after free Pu18, SPNs, and STPNs administration at different time points post-intravenous injection. **b** $T_1$-weighted MR images of orthotopic GBC-bearing nude mouse before and 1 h after intravenous injection of STPNs (top); and IVIS images of tumor-bearing mice depicting tumor bioluminescence (BL) and Pu18 fluorescence (FL) after STPNs administration at 1 h post-intravenous injection. **c** Healthy mice were intravenously injected once a week for a total of 5 times with 150 µL PBS or STPNs (irradiated by 980 nm laser source before injection at a power density of 2.0 W cm$^{-2}$ for 5 min) and killed at day 40 for hematological analysis ($n = 3$ independent experiments). **d** Healthy mice were intravenously injected once a week for a total of 5 times with 150 µL PBS or STPNs (irradiated by 980 nm laser source before injection at a power density of 2.0 W cm$^{-2}$ for 5 min) and killed at day 40 for blood biochemical analysis ($n = 3$ independent experiments). The data are represented as mean ± SD ($n = 3$ mice per group).

section analysis of vital organs (heart, lung, liver, kidney, and spleen) also showed no apparent differences between treated and untreated mice (Supplementary Fig. 35).

## Discussion

Phototherapy of deep tumors is hampered by ineffective tumor penetration, low tumor selectivity, and high phototoxicity of phototherapeutic agents. To address these issues, we developed acidic TME- responsive STPNs for HER2-targeted GBC therapy. In our design, we emphasize the ability of STPNs to be pre-activated by 980 nm laser irradiation prior to intravenous administration. At neutral blood pH of ~7.4, the photosensitizers within the STPNs are self-quenched and lack detectable photoactivity. As confirmed by the DPBF $^1O_2$ indicator, the DPBF absorption of STPNs at pH 7.4 decreased minimally in response to NIR irradiation, indicating that insignificant amount of ROS was generated. This demonstrated the biosafety of pre-activated STPNs at physiological pH. Upon reaching the tumor site, STPNs are disassembled into isolated UCNPs due to the acidic nature of the TME[38], leading to the recovery of Pu18 photoactivity and the production of ROS for antitumor therapy. Furthermore, we evaluated the afterglow intensity of STPNs under different power and irradiation time. As depicted in Supplementary Fig. 36, the afterglow intensity increases as irradiation power and duration increase. Interestingly, as the laser irradiation time increases at a power density of 4.0 W cm$^{-2}$, the afterglow intensity weakens. This is perhaps due to the intense power that leads to the disintegration of the nanoparticle structure[31]. Therefore, based on these results and the accumulation of STPNs at the tumor site within 1 h, STPNs were exposed to a 980 nm laser source at a power density of 2.0 W cm$^{-2}$ for 5 min. Moreover, STPNs can also be used as a contrast agent for MRI due to the Gd content within the UCNPs. Finally, the antitumor ability of STPNs for deep tumor treatment was evaluated using GBC cell lines and animal models. The results demonstrate the exceptional antitumor properties of STPNs with high therapeutic selectivity and excellent deep-tissue penetration both in vitro and in vivo.

In addition, TZB (HER2 antibody) on the STPNs can further combine with the generated ROS for a synergistic effect to enhance the antitumor efficiency. HER2 amplification/overexpression has been reported to be a prevalent cancer driver present in 15% of all cases of biliary tract cancers, including GBC[39,40]. Genomic profiling has identified HER2 as a druggable molecular target, and its monoclonal antibodies, such as TZB and zanidatamab, have demonstrated excellent clinical efficacy in HER2-positive GBC[40–42]. In most HER2-positive tumors, elevated HER2 regulates PI3K-Akt signaling to activate the nuclear translocation of the NFκB-p65/p50 complex, resulting in the release of inflammatory cytokines and growth factors that inhibit apoptosis and promote tumor proliferation, angiogenesis, and tumor invasion[35,43–45]. However, HER2 antibody resistance and cardiotoxicity pose clinical challenges. Therefore, combining HER2 antibodies with additional agents may be a promising strategy[39,46,47]. ROS-induced apoptosis by interfering with cellular signaling events has emerged as a viable method for the treatment of cancer[33], and the NFκB signaling pathway plays a critical role in this effect[34]. When higher levels of ROS are generated in the cell, overexpression of the antioxidant protein TRX1 decreases NF-κB nuclear translocation by inhibiting IκB degradation[48]. As a result, ROS generated by STPNs after 980 nm laser irradiation can induce apoptosis in tumor cells via numerous signaling pathways, including NFκB inhibition. In addition, TZB on the STPNs can bind to HER2, which is abundantly expressed on the cell membrane, to inhibit nuclear translocation, thereby enhancing tumor cell apoptosis. Due to the synergistic effect between TZB and ROS, a lower concentration of TZB can serve as a targeting agent and a potent antitumor agent for STPNs. Additionally, STPNs can also be extended to treat other cancers by simply changing the corresponding antibody,

which is another invaluable element in the clinical translation of lanthanide-doped UCNPs for deep tumor phototherapy.

The unique design of STPNs to be pre-activated prior to systemic administration has potential clinical impact. To date, the clinical application of phototherapy for deep tumor treatment has been restricted by the limited tissue penetration depth of the red or NIR light to activate the photosensitizers. Since living tissue contains chromophores that strongly absorb visible light[49], the red light used for PDT typically penetrates only 1–3 mm into biological tissue[50]. For deeper light penetration, the irradiated beam should be at the optical window of the tissue in the NIR range between 700 and 1300 nm[51]. Nevertheless, NIR irradiation can only penetrate up to 3–4 cm of the tissue thickness[52]. Currently, most research regarding lanthanide-doped UCNPs for cancer PDT uses 980 nm NIR irradiation that corresponds to the absorption of Yb$^{3+}$, a commonly used sensitizer. Although several studies have developed various 980 nm-triggered nanoparticles for deep tumor PDT[53,54], most of these nanomaterials are only feasible for superficial skin penetration such as breast cancer. The treatment of deep tumors such as GBC, liver cancer, pancreatic cancer, colon cancer, and so on, has an impenetrable depth of over 4 cm, which is an impossible barrier for any NIR light to penetrate. It has been reported that 980 nm NIR light attenuates while passing through tissues, significantly reducing the penetration depth. In addition, the high absorption of water at around 970 nm could contribute to the heating of the surrounding tissue[20,55], which is highly impractical for clinical applications. Therefore, the unique ability of STPNs to be pre-activated with 980 nm laser irradiation prior to intravenous administration may circumvent these obstacles to effectively treat deep tumors.

## Methods

### Ethical regulations

All animal handling protocols and experiments were approved by the Guidelines for Care and Use of Laboratory Animals of Zhejiang University (Protocol No. 24417). In accordance with the requirements of the Laboratory Animal Welfare and Ethics Committee of Zhejiang University, the size of the subcutaneous tumor and body tumor of mice must not exceed 1000 mm$^3$, in which the diameter of any dimension must be <10 mm. Once this size is reached, euthanasia must be performed. In every animal experiment described in this article, the maximal tumor size/burden of the mouse was never exceeded.

### Materials

All precursors and solvents were obtained commercially and used without further purification. L-glutamic acid γ-benzyl ester (BLG), bis-(trichloromethyl)-carbonate (triphosgene), hexane, tetrahydrofuran (THF), *N,N*-dimethylformamide (DMF), dichloromethane (DCM), dichloroacetic acid (DCA), hydrogen bromide (33 wt%) in acetic acid, dicyclohexylcarbodiimide (DCC), *N*-(3-dimethylaminopropyl)-*N*′-ethylcarbodiimide hydrochloride (EDC), *N*-hydroxysuccinimide (NHS), dimethyl sulfoxide (DMSO), 3-phenyl-1-propylamine (PPA), 1-(3-aminopropyl) imidazole (API), triethylamine, GdCl$_3$·6H$_2$O (99.9%), YbCl$_3$·6H$_2$O (99.99%), TmCl$_3$·6H$_2$O (99.99%), Ca(CH$_3$COO)$_2$·H$_2$O (99.0%), Eu(CH$_3$COO)$_3$·xH$_2$O (99.99%), oleic acid (OA), 1-octadecene (ODE), cyclohexane, sodium hydroxide, ammonium fluoride, methanol, 1,3-diphenylisobenzofuran (DPBF), acetonitrile, purine, and 5,5-dimethyl-1-pyrroline N-oxide (DMPO) were purchased from Aladdin Co. (China). Diethyl ether, hydrochloric acid (HCl), and ethanol were purchased from Sinopharm Co. (China). Amine-poly(ethylene glycol)-carboxyl (NH$_2$-PEG$_{2K}$-COOH) and Pluronic F-127 were purchased from Macklin Co. (China). Purpurin 18 (Pu18) and Trastuzumab (TZB) were purchased from MCE Co. (USA).

### γ-Benzyl-L-glutamate-*N*-carboxyanhydride synthesis

The synthesis of γ-Benzyl-L-glutamate-*N*-carboxyanhydride (BLG-NCA) was modified from previous work[56]. Briefly, L-glutamic acid γ-benzyl

ester (BLG) (14.0 g, 59.0 mmol) and triphosgene (20.0 g, 67.4 mmol) were mixed in anhydrous tetrahydrofuran (THF) (200 mL) and stirred at 50 °C for 2 h under argon gas. The reaction mixture was cooled and filtered at room temperature. The product was then purified via precipitation with cool hexane (500 mL) three times, collected by filtration, and dried under vacuum. The resulting BLG-NCA is obtained as a white fluffy powder.

## SPLs synthesis

SPLs were synthesized using carboxyl-poly(ethylene glycol)-block-poly(benzyl-L-glutamate) (COOH-PEG-PBLG) as a template (Supplementary Fig. 1). To prepare COOH-PEG-PBLG, BLG-NCA (3.156 g, 12.0 mmol) ring-opening polymerization was initiated in a mixture of $N,N$-dimethylformamide (DMF) (20 mL) and dichloromethane (DCM) (50 mL) with the primary amino group of amine-poly(ethylene glycol)-carboxyl (NH$_2$-PEG$_{2K}$-COOH) (molecular weight = 2000 Da, 0.24 g, 0.12 mmol) at 40 °C for 48 h under argon gas. COOH-PEG-PBLG was then purified via precipitation in excess of diethyl ether three times. Next, COOH-PEG-PLG was obtained by the deprotection of PBLG. COOH-PEG-PBLG (1.0 g, 57.35 μmol) was dissolved in dichloroacetic acid (DCA) (50 mL) and HBr/acetic acid (33 wt%, 15 mL) was added and stirred at 30 °C for 2 h under argon gas. The product was obtained via precipitation in excess of diethyl ether three times and dried under vacuum. After dissolving the product in DMF, the solution was dialyzed against deionized water for 72 h (MWCO: 7000 Da) and lyophilized to obtain COOH-PEG-PLG. To synthesize COOH-PEG-PLG-Pu18, Purpurin 18 (Pu18) was attached to the amine groups of COOH-PEG-PLG through the conventional carbodiimide reaction. COOH-PEG-PLG (0.5 g, 44.91 μmol) and a mixture of Pu18 (25.4 mg, 44.91 μmol), dicyclohexylcarbodiimide (DCC) (11.1 mg, 53.89 μmol), and $N$-hydroxysuccinimide (NHS) (6.2 mg, 53.89 μmol) were separately dissolved in dimethyl sulfoxide (DMSO) (5 mL) and stirred for 3 h at room temperature before the condensation reaction. The two solutions were then mixed and stirred for 24 h at room temperature under argon gas. Finally, to eliminate insoluble by-products, the reaction mixture was filtered, dialyzed against deionized water for 48 h (MWCO: 1000 Da), and lyophilized to obtain COOH-PEG-PLG-Pu18. COOH-PEG-p(API/PPA)-Pu18 was synthesized through platform ligand aminolysis using 3-phenyl-1-propylamine (PPA) and 1-(3-aminopropyl) imidazole (API). COOH-PEG-PLG-Pu18 (0.25 g, 13.89 μmol) was dissolved in DMSO (5 mL) and NHS (111.9 mg, 972.3 μmol) and $N$-(3-dimethylaminopropyl)-$N'$-ethylcarbodiimide hydrochloride (EDC) (186.4 mg, 972.3 μmol) was added and stirred at room temperature under argon gas. After 1 h, PPA (65.6 mg, 486.15 μmol) was then added to the reaction mixture and stirred at room temperature for another 1 h under argon gas. Next, API (60.8 mg, 486.15 μmol) was added and continued stirring at room temperature for 24 h under argon gas. Finally, the reaction mixture was then added dropwise to cool hydrochloric acid (HCl) solution (0.1 M, 20 mL), dialyzed against 0.01 M HCl aqueous solution (MWCO: 1000 Da), and lyophilized to obtain COOH-PEG-p(API/PPA)-Pu18.

## NaGdF$_4$:Yb,Tm@CaF$_2$:Eu UCNPs synthesis

The synthesis of UCNPs was modified from previous work[24]. To synthesize NaGdF$_4$:Yb,Tm, GdCl$_3$·6H$_2$O (185.6 mg, 0.50 mmol), YbCl$_3$·6H$_2$O (189.9 mg, 0.49 mmol), and TmCl$_3$·6H$_2$O (3.8 mg, 0.01 mmol) was added into a three-neck flask containing oleic acid (OA) (6 mL) and 1-octadecene (ODE) (15 mL). The solution was heated to 150 °C for 40 min under argon gas (alternate cycles of argon aeration and vacuum) to remove volatile liquids. After a clear solution forms, decrease the temperature to 50 °C and slowly add 10 mL methanol solution containing sodium hydroxide (100.0 mg, 2.50 mmol) and ammonium fluoride (148.2 mg, 4.00 mmol) to the reaction solution and continue stirring for 40 min under argon atmosphere. Next, heat the solution to 100 °C for 30 min (alternate cycles of

argon aeration and vacuum) to remove the methanol, then to 300 °C for 90 min under argon gas. After the reaction, the reaction mixture was cooled to room temperature, precipitated with ethanol, followed by washing with ethanol three times, and dispersed in cyclohexane (10 mL). To synthesize NaGdF$_4$:Yb,Tm@CaF$_2$:Eu, Ca(CH$_3$COO)$_2$·H$_2$O (149.8 mg, 0.85 mmol) and Eu(CH$_3$COO)$_3$·xH$_2$O (49.4 mg, 0.15 mmol) was added into a three-neck flask containing OA (6 mL) and ODE (15 mL). The solution was heated to 150 °C for 40 min under argon gas (alternate cycles of argon aeration and vacuum) to remove volatile liquids. After a clear solution forms, decrease the temperature to 50 °C and slowly add NaGdF$_4$:Yb,Tm (0.5 mmol) to the reaction solution and continue stirring for 30 min under argon gas. Next, slowly add 10 mL methanol solution containing sodium hydroxide (100 mg, 2.5 mmol) and ammonium fluoride (148.2 mg, 4.0 mmol) to the reaction solution and continue stirring for 40 min under argon gas. Heat the solution to 100 °C for 30 min (alternate cycles of argon aeration and vacuum) to remove the methanol, then to 300 °C for 90 min under argon gas. After the reaction, the reaction mixture was cooled to room temperature, precipitated with ethanol, followed by washing with ethanol three times, and finally dispersed in cyclohexane (10 mL).

## STPNs synthesis

COOH-PEG-p(API/PPA)-Pu18 (20 mg, 1.02 μmol) was dissolved in deionized water (10 mL) and UCNPs solution (in cyclohexane; 200 μL, 10 mg/mL) was slowly added into the solution under an ultrasonic condition to induce UCNP encapsulation and micelle formation. The reaction mixture was then stirred for 1 h at room temperature before removing cyclohexane by rotary evaporation at 60 °C. Pluronic F-127 (8 mg) was then added, followed by adjusting the pH value to 7.4 with sodium hydroxide, and continued stirring at room temperature for 1 h. The excess ligands were removed with a spin filter (Millipore; MWCO: 100 kDa, 10,000 × $g$, 10 min) and dispersed in PBS (5 mL). Next, EDC (2.9 mg, 15.0 μmol) and NHS (1.7 mg, 15.0 μmol) in 2 mL PBS were added and stirred at room temperature under argon gas. After 1 h, a solution of trastuzumab (TZB) (1 mg) in 3 mL PBS was added, followed by adjusting the pH value to 8 with triethylamine, and continued stirring at 4 °C for 12 h under argon gas. Finally, the excess ligands were removed with a spin filter (MWCO: 100 kDa, 10,000 g, 10 min) and redispersed in PBS. SITPNs and SPNs were synthesized using the same process, but with ligands without imidazole groups or TZB conjugation, respectively.

## Characterization

Transmission electron microscopy (TEM) was taken with HT-7700 (Hitachi, Japan) at 120 kV for morphology characterization. High-angle annular dark-field scanning transmission electron microscopy (HADFF-STEM) imaging and mapping were conducted on Tecnai G2 F20 S-Twin (FEI, USA). Zeta potentials and dynamic light scattering (DLS) were conducted on Zetasizer Nano ZS90 (Malvern Instruments, UK). $^1$H-NMR analysis was conducted on Ascend 600 (Bruker, Swiss) using DMSO-$d_6$ as the solvent. FTIR analysis was conducted on Nicolet AVA TAR370 (ThermoFisher, USA). UCL emission spectra were conducted on FLSP920 (Edinburgh Instruments, UK). Absorption spectrum was conducted on Shimadzu UV-1900i (Shimadzu, Japan). ESR measurements were conducted on A300 Spectrometer (Bruker, Swiss), using a 1.0 mm quartz capillary. X-ray diffraction (XRD) patterns were conducted on D8 ADVANCE (Bruker, Swiss). NIR light source was supplied by an optical fiber-coupled 980 nm diode laser (STL980T1-7W, Radium Laser Co., China).

## STPNs transmittance analysis at different pH

The light-transmittance of STPNs solutions (2 mg/mL) was conducted using a UV-vis spectrophotometer at 500 nm with solution pH adjustments from 7.5 to 4.0 by adding HCl (0.1 M).

## Fluorescence intensity analysis at different pH

The fluorescence intensity of STPNs or SITPNs solutions (2 mg/mL) was conducted using a fluorescence spectrophotometer at 400 nm excitation with solution pH adjustments from 7.4 to 5.5 by adding HCl (0.1 M).

## Critical aggregation concentration analysis

Pyrene ($1.0 \times 10^{-5}$ M) was mixed in methanol, placed in a test tube, and distilled at 60 °C under vacuum to remove the methanol. Varying concentrations of COOH-PEG-p(API/PPA)-Pu18 solution were added and ultrasonicated for 30 min, with the ultimate pyrene concentration in each sample being $1.0 \times 10^{-7}$ M. The fluorescence intensity was analyzed using a fluorescence spectrophotometer at excitation and emission wavelengths of 337 nm and 375 nm, respectively. The point whereby emission intensity declines sharply reflects the critical aggregation concentration.

## High-performance liquid chromatography analysis

High-performance liquid chromatography analysis was conducted on Agilent 1200 (Marshall Scientific, USA). For TZB, the separation was performed using Jupiter 5 μm C18 300 Å LC column (250 mm×4.6 mm) under the following gradient. Solvent A: acetonitrile. Solvent B: 0.1% formic acid. The gradient was mixed from solvent A and B: 0–15 min, 10% A; 15–30 min, 30% A; 30–32 min, 45% A; 32–36 min, 90% A; and 36–48 min, 10% A. The volume injected was 20.0 μL at a flow rate of 1.0 mL/min. TZB were monitored using FLD, with an excitation wavelength of 278 nm and an emission wavelength of 343 nm. For Pu18, the separation was performed using Eclipse XD8-C18 5 μm column (250 mm × 4.6 mm) under the following gradient. Solvent A: acetonitrile. Solvent B: 0.1% trifluoroacetic acid. The gradient was mixed from solvent A and B: 0–15 min, 70% A; and 15–20 min, 100% A. The volume injected was 20.0 μL at a flow rate of 1.0 mL/min. Pu18 was monitored using a UV detector at 410 nm.

## Singlet oxygen generation analysis via DPBF

For pH-dependent DPBF absorbance, DPBF was mixed with acetonitrile (20 μL, 8 mM), added into STPNs solutions (2 mg/mL, 3 mL) at different pH and was irradiated with 980 nm laser source (2.0 W cm$^{-2}$) at the specified time intervals. For time course $^1O_2$ generation, DPBF was mixed with acetonitrile (20 μL, 8 mM), added into STPNs solutions (pH 5.5; 2 mg/mL, 3 mL), and was irradiated with 980 or 650 nm laser source (2.0 W cm$^{-2}$) for 5 min. The absorption spectra were then measured using a UV-vis spectrophotometer. During laser irradiation, all samples were mixed and stirred to ensure the dispersion of light energy throughout the sample. All experiments were performed in triplicate.

## Determination of $^1O_2$ quantum yield

STPNs $^1O_2$ quantum yield is determined by referring to a reference, methylene blue in dichloromethane, as 57%[57,58]. The relative quantum yields were obtained according to the following equation:

$$\Phi_s = \Phi_{ref} \cdot \frac{k_s}{k_{ref}} \cdot \frac{F_{ref}}{F_s} \cdot \frac{PF_{ref}}{PF_s} \qquad (1)$$

Where $\Phi_s$ and $\Phi_{ref}$ designate $^1O_2$ quantum yield of STPNs and reference methylene blue (ref). $k$ is the slope of time-dependent decrease in DPBF absorbance, $F$ is the absorption correction factor ($F = 1 - 10^{-OD}$, $OD$ is the absorbance at the irradiation wavelength), and $PF$ is absorbed photonic flux.

## Singlet oxygen detection via electron spin resonance

ESR spectra were utilized to detect ROS production from STPNs. STPNs solutions (150 μg/mL, 200 μL) were mixed with 5,5-dimethyl-1-pyrroline N-oxide (DMPO) (5.0 mmol, 100 μL) and irradiated with 980 nm laser source (2.0 W cm$^{-2}$). The reaction mixture was then injected into the quartz capillary for ESR analysis.

## Cell culture

Human gallbladder carcinoma cell line GBC-SD, human cholangiocarcinoma cell line RBE, HuCCT1, CCLP1, and human gallbladder epithelial cell line HGBEC were obtained from the Shanghai Institute for Biological Science, Chinese Academy of Science (Shanghai, China); Human gallbladder carcinoma cell lines NOZ, EH-GB1, SGC-996, and human intrahepatic biliary epithelial cell line HIBEC were supplied by Dr. Ying-Bin Liu's lab at Xinhua Hospital Affiliated to Shanghai Jiao Tong University School of Medicine, China. RBE, HuCCT1, CCLP1, NOZ, and EH-GB1 were cultivated at Dulbecco's Modified Eagle Medium (DMEM) supplemented with 10% fetal bovine serum (FBS) and 1% penicillin/streptomycin at 37 °C in 5% CO$_2$. GBC-SD, HGBEC, SGC-996, and HIBEC were cultivated at Roswell Park Memorial Institute (RPMI) 1640 supplemented with 10% fetal bovine serum (FBS) and 1% penicillin/streptomycin at 37 °C in 5% CO$_2$.

## Cellular uptake

GBC-SD and EH-GB1 cells were exposed to STPNs at tumor microenvironment pH 6.5 and physiological pH 7.4 to validate uptake. GBC-SD cells ($5 \times 10^4$/mL) and EH-GB1 cells ($5 \times 10^4$/mL) were seeded in 500 μL medium in 24-well cell culture plates, incubated overnight, and STPNs (0.4 mg/mL) were then added to the plates. After 0.5 h, 1 h, and 4 h of incubation, the cells were then washed with PBS three times. The samples were fixed with 4% paraformaldehyde for 10 min and incubated with DAPI (0.01%) for 10 min. Cellular uptake was monitored by fluorescent microscopy EVOS FL Auto 2 (Thermo Fisher Scientific, USA) at different time points.

## Cryo-TEM cell imaging

GBC-SD and EH-GB1 cells were exposed to STPNs at tumorous microenvironment pH 6.5 to validate uptake. GBC-SD cells ($1.5 \times 10^6$/mL) and EH-GB1 cells ($1.5 \times 10^6$/mL) were seeded in 6 cm Petri dishes until adherent and replaced with STPNs (0.5 mg/mL). After incubation for 0.5 h, 1 h, and 4 h, the cells were first fixed with 2.5% glutaraldehyde in PBS (0.1 M, pH 7.0) overnight, washed three times with PBS, and subsequently post-fixed with 1% OsO$_4$ in PBS for 1 h and washed three times with PBS. Next, the cell samples were dehydrated by a graded series of ethanol (30%, 50%, 70%, 80%) and acetone (90%, 95%, 100%) for 15 min. The cell samples were then placed in a mixture of 1:1 and 1:3 acetone and Spurr resin mixture for 1 h and 3 h respectively and transferred to the final Spurr resin mixture overnight. Finally, the specimens were placed in Spurr resin and heated to 70 °C for 9 h and were sectioned into ultrathin slices with LEICA EM UC7. The sections were then stained with alkaline lead citrate and uranyl acetate for 10 min. The morphology was observed using TEM microscopy Talos L120C (Thermo Fisher Scientific, USA) at 120 kV.

## Cell Counting Kit-8 assay

The in vitro cytotoxicity of nanoparticles was measured using the CCK-8 (Yeasen, Shanghai) assay according to the manufacturer's instructions. GBC-SD, EH-GB1 cells were seeded into 96-well cell culture plates at $0.6 \times 10^4$ per well until adherent and replaced with TZB in different concentrations (3.16–1000 nM) or STPNs (3.16–1000 nM, TZB concentration in STPNs) and cultured for 48 h. The absorbance was measured by a microplate reader (Thermo Fisher Scientific, USA) at 450 nm. All experiments were performed in triplicate.

Cell damage caused by ROS production from Pu18 in nanoparticles under laser irradiation was measured via CCK-8 assay. GBC-SD and EH-GB1 cells were seeded into 96-well cell culture plates at $0.6 \times 10^4$ per well until adherent and replaced with SPNs loaded with different concentrations of Pu18 (0.65–65.56 μg/mL, Pu18 concentration in SPNs) for 12 h incubation. After removing the nanoparticles,

cells were transferred to a new media and irradiated with a 980 nm laser source at a power density of $2.0\,W\,cm^{-2}$ for 5 min with a 1 min interval. The cells were incubated at 37 °C for an additional 36 h before measuring on a microplate reader at 450 nm to evaluate their viability compared to the unirradiated cells. All experiments were performed in triplicate.

Finally, the effect of nanoparticles on GBC cell proliferation was also measured by CCK-8 and nanoparticles cell proliferation assay. GBC-SD and EH-GB1 cells were seeded in 96-well plates at $0.6 \times 10^4$ per well until adherent and replaced with different media for 12 h incubation. Each cell line was divided into 8 groups: (1) NC (normal media), (2) NC (normal media) with laser irradiation, (3) TZB (50 nM), (4) TZB (50 nM) with laser irradiation, (5) SPNs (the same nanoparticle concentration as group 7), (6) SPNs (the same nanoparticle concentration as group 7) with laser irradiation, (7) STPNs (50 nM, TZB concentration in STPNs), and (8) STPNs (50 nM, TZB concentration in STPNs) with laser irradiation. After the removal of old media, cells were transferred into fresh media, and groups 2, 4, 6, and 8 were irradiated by a 980 nm laser source at a power density of $2.0\,W\,cm^{-2}$ for 5 min with 1 min intervals. The cells were then incubated at 37 °C for an additional 36 h before measuring on a microplate reader at 450 nm to determine their viabilities relative to the control unirradiated cells. All experiments were performed in triplicate.

## Western blotting
To detect HER2 expression, HIBEC, RBE, HuccT1, CCLP1, HGBEC, SGC-996, NOZ, GBC-SD, and EH-GB1 cells were seeded in 6-well plates at the density of $4 \times 10^5$ per well. After 48 h, the total protein of each cell line was extracted. For mechanism validation, GBC-SD and EH-GB1 cells were seeded in 6-well plates at the density of $4 \times 10^5$ per well until adherent and replaced with different media. Each cell line was also divided into 8 groups and treated as previously described for the cell proliferation assay. After 48 h, total protein was extracted from each cell group. Then western blotting was performed according to the standard methods. The following antibodies, Rabbit monoclonal anti-HER2 (Abcam, Catalog no. ab134182, 1:1000 dilution), Mouse monoclonal anti-GAPDH (Abcam, Catalog no. ab8245, 1:5000 dilution), Rabbit monoclonal anti-NFκB p65 (Abcam, Catalog no. ab32536, 1:10,000 dilution), Rabbit monoclonal anti-NFκB p50 (Abcam, Catalog no. ab32360, 1:10,000 dilution), Mouse monoclonal anti-beta actin (Abcam, Catalog no. ab8226, Use a concentration of $1\,\mu g/ml$), and Rabbit polyclonal anti-Lamin B1 (Abcam, Nuclear Envelope Marker, Catalog no. ab16048; Use a concentration of $0.1\,\mu g/ml$) were obtained from Abcam. Supplementary Figures 37 and 38 depicts the source data underlying Supplementary Fig. 26 and Fig. 4b, respectively.

## ROS assays
GBC-SD and EH-GB1 cells were seeded in 24-well plates at $1 \times 10^5$ per well until adherent and replaced with different media for 12 h incubation. Each cell line was also divided into 8 groups and treated similarly as mentioned previously. 4 h after irradiation, for flow cytometry analysis, adherent cells of all groups were harvested and washed with PBS 3 times; for observation of the fluorescence cell imager, adherent cells of all groups were washed with PBS 3 times. Next, each group was stained with $500\,\mu L$ ($25\,\mu M$) 2,7-dichlorodihydrofluorescein diacetate (Sigma-Aldrich, USA) for 30 min at 37 °C in the dark and washed by PBS 3 times. The cells were then immediately analyzed by flow cytometry (BD LSRFortessa™, USA) and observed by a fluorescence cell imager (Bio-Rad, USA), respectively. All experiments were performed in triplicate.

## PI/Annexin V apoptosis assay
GBC-SD and EH-GB1 cells were seeded in 24-well plates at $1 \times 10^5$ per well until adherent and replaced with different media for 12 h incubation. Each cell line was also divided into 8 groups and treated similarly

as mentioned previously. After 48 h, all cells in each group, including attached and floating cells, were collected through trypsinization (0.25% Trypsin) without EDTA (Gibco), then washed with PBS. Annexin V-FITC and PI were used to identify apoptotic cells using the Annexin V-FITC Apoptosis Detection Kit I (BD Biosciences) according to the manufacturer's instructions. Viable and dead cells were detected via a flow cytometer (BD LSRFortessa™, USA).

## Colony-forming assays
GBC-SD and EH-GB1 cells were seeded in 24-well plates at $1 \times 10^5$ per well until adherent and replaced with different media for 12 h incubation. Each cell line was also divided into 8 groups and treated similarly as mentioned previously. After 48 h, all 8 groups were seeded in 6-well plates at $1 \times 10^3$ per well. Next, each group was cultured for an additional week. Finally, each well was fixed with 4% paraformaldehyde and stained with 0.1% crystal violet according to the manufacturer's instructions, and a full visual was captured by a gel imager (Bio-Rad, USA). All experiments were performed in triplicate.

## Animals
Female BALB/c nude mice (4 weeks old) were obtained from Shanghai SLAC Laboratory Animal Co., Ltd. All animals were bred in a pathogen-free facility with a 12 h light/dark cycle at $20 \pm 3$ °C and 40–50% humidity and had ad libitum access to food and water.

## Orthotopic gallbladder cancer modeling
An orthotopic GBC-SD gallbladder cancer-bearing mouse model was established with a high success rate of nearly 90% by implanting minced gallbladder cancer tissue into the left lobe of the liver in BALB/c nude mice. GBC-SD-Luc cells ($5 \times 10^6$ cells) in $100\,\mu L$ PBS were subcutaneously injected into the right armpit of each female BALB/c nude mouse. When the tumor volume reached $800\,mm^3$, mice were monitored by bioluminescence using an imaging system (PerkinElmer IVIS® Lumina LT, USA) 10 min after the mice were anesthetized and injected with D-luciferin sodium salt ($15\,mg\,mL^{-1}$ in $1 \times$ DPBS) at $150\,mg\,kg^{-1}$. A mouse tumor with substantial fluorescence intensity was selected, diced into $0.5\,mm \times 0.5\,mm \times 0.5\,mm$ pieces, and washed with PBS for orthotopic inoculation preparation. Subsequently, other female BALB/c nude mice were operated on with sterilized instruments, and aseptic principles were maintained. Under the xiphoid process of the mouse, a 1 cm incision was created, and the left liver lobe was extruded via the abdominal orifice. Using the tip of the ophthalmic forceps (XinGong, Chian), the liver leaf was removed by creating a 3mm-long and 3mm-deep channel on its surface. To stop the bleeding, apply pressure to the wound using the sterile cotton swab. The prepared tumor fragment was then placed into the channel on the surface of the liver, and the small opening was compressed for 1 to 2 min using a cotton swab. After the bleeding is stopped, the left lobe of the liver was carefully reinserted into the abdominal cavity, and the abdominal opening was closed layer by layer. As previously mentioned, all mice were monitored with bioluminescence 10 days after surgery. All mice exhibiting detectable fluorescence intensity in the abdomen were regarded as successful models. The mouse was euthanatized when its tumor exceeded the pre-specified maximal tumor volume of $1\,cm^3$ or when its maximum weight loss exceeded 20%, as required by animal ethics.

## Tumor inhibition experiment
For treatment, model mice were randomly divided into 4 groups ($n = 8$, 5 for monitoring survival and 3 for histological analysis): (1) PBS, (2) TZB (2.5 mg/kg, equivalent to the dose loaded in STPNs), (3) STPNs (115 mg/kg), and (4) STPNs (115 mg/kg) with NIR (irradiated by 980 nm laser source before injection at a power density of 2.0 W cm-2 for 5 min). TZB dosage falls within the range of clinically attainable levels. The mice were intravenously injected with $150\,\mu L$ of the respective formulations and monitored by bioluminescence as described above

once a week for five times. The mouse's body weight was measured every four days. After 38 days, all the mice were monitored for survival assessment. Meanwhile, orthotopic tumor tissue was removed from one mouse in each treatment group, fixed in 4% paraformaldehyde, and embedded in paraffin. Paraffin-embedded tissue slices then underwent H&E and Ki-67-antigen.

### Afterglow and fluorescence luminescence imaging

Afterglow and fluorescence imaging were performed using an IVIS Spectrum imaging system (PerkinElmer IVIS® Lumina LT, USA) in bioluminescence and fluorescence modes, respectively. For afterglow, 150 μL STPNs (15 mg/mL, the same nanoparticle content as the treatment group) in a tube were irradiated by a 980 nm laser source at a power density of $2.0 \, W \, cm^{-2}$ for 5 min. In vitro afterglow signals were collected every 120 s for a total of 2 h with an open filter. The afterglow spectra were collected for 30 s with specific emission filters. The afterglow intensities were quantified by measuring the signal intensity of the region of interest (ROI) using Living Imaging 4.7.3 Software. For fluorescence imaging, tumor-bearing model mice received separate intravenous injections of free Pu18, SPNs, and STPNs (10 days after orthotopic GBC inoculation). Fluorescence images ($\lambda_{ex} = 700 \, nm$; $\lambda_{em} = 710 \, nm$) of the mice were acquired by the NIR imaging system at pre-injection and 0.5 h, 1 h, 2 h, 4 h, 12 h, and 24 h post-injection. Similarly, STPNs were administered by intravenous injection separately for tumor-bearing model mice. Next, the mice were killed at different time points (0.5, 1, 2, 4, 12, 24 h) after intravenous injection, with the heart, liver (the tumor remains inside), spleen, lung, and kidney excised for the observation of the biodistribution of nanoparticles via imaging.

### MR imaging

$NaGdF_4$:Yb,Tm@$CaF_2$:Eu UCNPs and STPNs with various concentrations (0–1.0 mM) were scanned under a 3.0 T MR system (Discovery 750w, GE Healthcare) at room temperature. For in vivo MR imaging, the selected tumor-bearing model mice received intravenous injections STPNs (24 days after orthotopic GBC inoculation). $T_1$-weighted MR images of the mouse were acquired at pre-injection and 1 h post-injection on a 3.0 T MR system equipped with an animal-imaging coil.

### Biosafety assessment

Normal female BALB/c nude mice at the age of 5–6 weeks were randomly divided into two groups ($n = 3$). Healthy mice were intravenously injected once a week for a total of 5 times with 150 μL STPNs solution (115 mg/kg, irradiated by 980 nm laser source before injection at a power density of $2.0 \, W \, cm^{-2}$ for 5 min), and healthy mice with PBS injection were used as the control group. For histopathological analyses of major organs, different groups of mice were killed on day 40 of treatment to collect the major organs (heart, liver, spleen, lung, and kidney). The tissue samples were fixed in a 4% paraformaldehyde solution, stained with hematoxylin and eosin, and then examined under a digital microscope. For blood analysis, complete blood investigation and serum biochemistry assays were carried out by collecting 600 μL of blood from the mice. The mean corpuscular hemoglobin (MCH), hemoglobin concentration (MCHC), mean corpuscular volume (MCV), platelets (PLT), red blood cells (RBC), and white blood cells (WBC) were measured. Blood biochemical examination parameters, including alanine aminotransferase (ALT), aspartate aminotransferase (AST), albumin (ALB), direct bilirubin (DBIL), cholinesterase (CHE1), serum albumin (AlbG), and total protein (TP), were also measured.

### Statistics and reproducibility

All the analysis data are given as mean ± SD. The results were analyzed by the Student's *t*-test between two groups. Other statistic information was provided accordingly in the figure legends. For the data in Figs. 2a; 3a, b; 4b, c; and 5e and Supplementary Fig. 35, three experiments were repeated independently with similar results, and results from representative experiments were shown.

### Reporting summary

Further information on research design is available in the Nature Portfolio Reporting Summary linked to this article.

## Data availability

All data generated or analyzed during this study are included in this published article (and its Supplementary Information files). All other data are available from the corresponding authors upon request.

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

## Acknowledgements

This work was supported by the National Natural Science Foundation of China (No. 82202873), the Natural Science Foundation of Zhejiang Province (No. LQ22H160003), the Fundamental Research Funds for the Central Universities (No. 226-2022-00141), and the National Key R&D Program of China (No. 2021YFA0909900). The authors would like to thank Chaogang Xing, Qin Zhang, and Lijuan Mao from the Analysis Center of Agrobiology and Environmental Sciences, Zhejiang University; Yu Liu and Xinhang Jiang from the College of Life Sciences, Zhejiang University; Fang Chen from the Department of Chemistry, Zhejiang University; and Guoqing Zhu and Qingyun Lin from the Center for Electron Microscopy of Zhejiang University for their technical assistance in nanoparticle characterization; as well as Yuchen Zhang from the Center of Cryo-Electron Microscopy (CCEM), Zhejiang University for technical assistance on Cryo-TEM; and Prof. Yifeng Han from Zhejiang Sci-Tech University and Dr. Yiyuan Zhu from Zhejiang University for technical consultation. The figures in this article were created using Adobe Illustrator, Adobe Photoshop, BioRender, and Microsoft PowerPoint.

## Author contributions

S.J., S.L., T.Y., W.T., and M.C. designed the experiments, analyzed the data, and wrote the manuscript. W.T., Z.G., X.C., and M.C. supervised the project and revised the manuscript. S.J., T.Y., T.X., J.C., J.L., Z.L., J.Y., and Y.W. synthesized and characterized the nanoparticles. S.L., Y.S., T.C., B.Z., J.C., and J.H. performed in vitro and in vivo experiments. All authors discussed the results throughout the project and approved the final version of the manuscript.

## Competing interests

Z.G. is a scientific co-founder of ZenCapsule, Inc. The remaining authors declare that there are no conflicts of interest.
