## [Peer Review File · Nature Communications]

Reviewers' Comments:

Reviewer #1:

Remarks to the Author:

In this manuscript, the authors constructed stimuli sensitive tumor-targeted photodynamic nanoparticles (STPNs) with persistent luminescence for the treatment of deep tumors through a self-assembly method. In particular, they demonstrated antitumor therapeutic effects of STPNs in vivo. In my opinion, it is an interest work. Therefore, I recommend that this manuscript can be accepted for publication in nature communication after minor revision. My comments are given as follows.

1. Please characterize the morphology of CaF₂ shell and give its thickness.
2. Please discuss the upconversion luminescent process of NaGdF₄:Yb,Tm@CaF₂:Eu UCNPs and illustrate the role of Eu dopant in this process.
3. In Fig. 2j, a phosphorescent spectrum of STPNs recorded at 0.5 h or 1h after stopping 980 nm irradiation should be provided.
4. The ¹O₂ quantum yield of STPNs should be evaluated within 30 min after stopping 980 nm irradiation.

Reviewer #2:

Remarks to the Author:

This manuscript discusses a potential application of stimuli-sensitive tumor-targeted photodynamic nanoparticles (STPNs) with persistent luminescence for the treatment of deep tumors. According to the authors, photodynamic therapy (PDT) could be an effective diagnostic and therapeutic strategy for treating gallbladder cancer (GBC) by replacing conventional chemotherapy and radiotherapy which are unable to prolong the survival of patients with recurrent or advanced GBC. To address this problem the authors developed stimuli-sensitive tumor-targeted photodynamic nanoparticles (STPNs) consisting of the lanthanide ion-doped up-conversion nanoparticles, stimuli-responsive polymeric ligands, and trastuzumab, for HER2-targeted GBC therapy. Synthesis of such nanoparticles is certainly a nice achievement and demonstrates the outstanding chemistry qualifications of the authors. The idea of the work is that STPNs can be excited by 980 nm laser irradiation prior to intravenous administration, and upon reaching the tumor, due to the acidic environment, STPNs disassemble into isolated up-conversion nanoparticles which start to emit light due to persistent luminescence. The clever idea here is that light penetration depth is not a problem anymore as the light is generated by the PDT agent locally.

However, there are several caveats which make the proposed approach somewhat questionable. A decade ago photodynamic therapy was employed in a small study for treating deep lying prostate tumor in humans with limited success. In that study histopathological analysis of tissue biopsies taken six months post-PDT treatment showed there were still residual viable cancer cells present in the prostate tissue sections, and the authors hypothesized that the incomplete treatment of the prostate tissue could be due to an insufficient light power delivered to cancer cells. This conclusion emphasizes two important points: (a) development of a PDT modality which does not require the external illumination which is strongly attenuated by the overlaying tissue layers would be a significant advance, however (b) at the same time, sufficient power of the upconverted emission light is required for the PDT to be successful. A number of recent publications quote local light power of 1.5 mW·cm⁻² to 2.5 mW·cm⁻² as the lowest power required for PDT to work. Unfortunately, the manuscript does not provide the power of luminescent light achievable with the up-conversion nanoparticles. To overcome this problem and demonstrate that the approach works at least in the studies with the tumor-bearing mice, the authors indicate that tumor bioluminescence intensity in treated mice gradually decreased and disappeared. This is interpreted as an excellent tumor suppressive efficacy of the treatment. It is not entirely clear why bioluminescence intensity is used to demonstrate that PDT treatment works when an obvious gold standard would be the histopathological analysis of tissue of tumor-bearing and control mice after they are sacrificed. Moreover, it appears that mice were indeed sacrificed with H&E staining section analysis of vital organs showing no apparent differences between treated and untreated mice, posing obvious questions about the potential effect of the technique on tumor tissue. Also a

minor point is that the GBC is indeed very deadly but relatively rare cancer and the authors might consider more common cancers involving the gastrointestinal tract, for example, pancreatic cancer.

Overall, the manuscript is clearly written and presents an intriguing but early and not particularly complete results. The manuscript would certainly be of value for the community developing new nanoparticle-based photosensitizers for PDT. However, the work is not of the far-reaching scope or of ground-breaking value typical of manuscripts published in the Nature family journals and this manuscript is better suited for a specialized journal.

Reviewer #3:

Remarks to the Author:

The limited depth of light penetration largely hinders the practical use of PDT in vivo. Sarun et al. present a pH responsive, size switchable nanoplatfrom (STPN) for PDT/trastuzumab synergistic therapy in orthotopic gallbladder cancer. The afterglow luminescence of STPN excited NIR-laser served as a persistent light source for PDT, which overcomes the problem of tissue penetration of laser. The photoactivity of photosensitizer in the STPNs was partially quenched and recovered in the acidic tumor environment, thus reducing the toxicity during in vivo circulation. This work is an important extension of previously published work by the Ling group (*Adv. Mater.* 2018, 30, 1802808). I would recommend its publication in *Nature Communications*, after the following concerns are addressed?

(1) In Figure 2i, laser power should be given.

(2) Since the PDT is triggered by afterglow luminescence, the intensity and decay rate of afterglow luminescence is highly related with the final therapeutic outcome of PDT. The readers are more care about whether the afterglow luminescence is stronger or longer enough for effective PDT? It would be better to show more trials under various conditions (e.g. different laser power, different irradiation time)

(3) It would be beneficial to discuss the corresponding time points between afterglow luminescence decay and in vivo tumor accumulation of STPNs.

(4) Semi-quantitative analysis of western-blot in Figure 4b should be given.

Reviewer Comments to Author

Reviewer #1

Comments: *In this manuscript, the authors constructed stimuli sensitive tumor-targeted photodynamic nanoparticles (STPNs) with persistent luminescence for the treatment of deep tumors through a self-assembly method. In particular, they demonstrated antitumor therapeutic effects of STPNs in vivo. In my opinion, it is an interest work. Therefore, I recommend that this manuscript can be accepted for publication in nature communication after minor revision. My comments are given as follows.*

Response: Thank you very much for your kind comments. We have revised our manuscript based on your suggestions. We believe that this revision has significantly improved the quality of our manuscript.

Q1: *Please characterize the morphology of CaF₂ shell and give its thickness.*

Response: We have characterized and discussed the morphology of the CaF₂ shell in the results section of the manuscript (Supplementary Fig. 13). The thickness of the CaF₂ shell is 2.95 ± 0.53 nm.

Supplementary Figure 13: Characterization of CaF₂ shell morphology and thickness. a) HR-TEM imaging and b) HAADF-STEM imaging of NaGdF₄:Yb,Tm@CaF₂:Eu core@shell UCNPs; Inset: HAADF-STEM imaging of NaGdF₄:Yb,Tm@CaF₂:Eu core@shell UCNPs. c) CaF₂ shell thickness size distribution. d) NaGdF₄:Yb,Tm@CaF₂:Eu core@shell UCNPs EDS mapping of Gd (core) and Eu (shell).

Q2: Please discuss the upconversion luminescent process of NaGdF₄:Yb,Tm@CaF₂:Eu UCNPs and illustrate the role of Eu dopant in this process.

Response: The upconversion luminescent process of NaGdF₄:Yb,Tm@CaF₂:Eu UCNPs and the role of Eu dopant in this process have been discussed in the results section of the manuscript (Supplementary Fig. 16).

Supplementary Figure 16: a) UCL spectra and process of NaGdF₄:Yb,Tm@CaF₂:Eu core@shell UCNP and NaGdF₄:Yb,Tm core UCNP. b) Image of NaGdF₄:Yb,Tm@CaF₂:Eu core@shell UCNP (right) and NaGdF₄:Yb,Tm core UCNP (left) solutions in cyclohexane under 980 nm laser irradiation. c) The energy level configuration of NaGdF₄:Yb,Tm@CaF₂:Eu core@shell UCNPs.

Q3: In Fig. 2j, a phosphorescent spectrum of STPNs recorded at 0.5 h or 1h after stopping 980 nm irradiation should be provided.

Response: Due to the faint phosphorescent signal of porphyrins, current fluorescence spectroscopy instruments are unable to detect their phosphorescent spectrum. Therefore, the fluorescence images and afterglow luminescence spectra were captured on an IVIS Spectrum imaging system^{1,2}. We have provided STPNs fluorescence and afterglow images and intensity 5, 30, and 60 min after 980 nm laser irradiation; normalized fluorescence spectra of STPNs under 980 nm laser irradiation; and normalized afterglow luminescence spectra of STPNs 30 min after 980 nm laser irradiation in Supplementary Fig. 24.

Supplementary Figure 24: a) STPNs fluorescence and afterglow images and intensity 5, 30, and 60 min after 980 nm laser irradiation. b) Normalized fluorescence spectra of STPNs under 980 nm laser irradiation. c) Normalized afterglow luminescence spectra of STPNs 30 min after 980 nm laser irradiation. The data are represented as mean \pm SD (n=3).

Q4: The $^1\text{O}_2$ quantum yield of STPNs should be evaluated within 30 min after stopping 980 nm irradiation.

Response: Thank you for pointing this out. We have provided the $^1\text{O}_2$ quantum yield of STPNs 30 min after stopping 980 nm laser irradiation in the results section of the manuscript (Supplementary Fig. 23). STPNs $^1\text{O}_2$ quantum yield under continuous 980 nm laser irradiation is 8%, while the $^1\text{O}_2$ quantum yield 30 min after stopping 980 nm laser irradiation is 1%.

Supplementary Figure 23: a) STPNs decrease in DPBF absorbance under continuous 980 nm laser irradiation. b) STPNs absorption spectra at the laser irradiation wavelength region. c) Decrease in absorbance of DPBF in dichloromethane in the presence of reference photosensitizer methylene blue under continuous 660 nm laser irradiation. d) Time-dependent DPBF absorbance decrease at 414 nm in dichloromethane under the presence of STPNs and e) reference photosensitizer methylene blue. f) STPNs decrease in DPBF absorbance after 5 min of 980 nm laser irradiation. The data are represented as mean \pm SD (n=3).

Reviewer Comments to Author

Reviewer #2

Comments: *This manuscript discusses a potential application of stimuli-sensitive tumor-targeted photodynamic nanoparticles (STPNs) with persistent luminescence for the treatment of deep tumors. According to the authors, photodynamic therapy (PDT) could be an effective diagnostic and therapeutic strategy for treating gallbladder cancer (GBC) by replacing conventional chemotherapy and radiotherapy which are unable to prolong the survival of patients with recurrent or advanced GBC. To address this problem the authors developed stimuli-sensitive tumor-targeted photodynamic nanoparticles (STPNs) consisting of the lanthanide ion-doped up-conversion nanoparticles, stimuli-responsive polymeric ligands, and trastuzumab, for HER2-targeted GBC therapy. Synthesis of such nanoparticles is certainly a nice achievement and demonstrates the outstanding chemistry qualifications of the authors. The idea of the work is that STPNs can be excited by 980 nm laser irradiation prior to intravenous administration, and upon reaching the tumor; due to the acidic environment, STPNs disassemble into isolated up-conversion nanoparticles which start to emit light due to persistent luminescence. The clever idea here is that light penetration depth is not a problem anymore as the light is generated by the PDT agent locally.*

Response: Thank you very much for your suggestions. The manuscript was revised and elaborated as your comment suggested. We believe that this revision had made our manuscript much more coherent and comprehensible.

Comments: *However, there are several caveats which make the proposed approach somewhat questionable. A decade ago photodynamic therapy was employed in a small study for treating deep lying prostate tumor in humans with limited success. In that study histopathological analysis of tissue biopsies taken six months post-PDT treatment showed there were still residual viable cancer cells present in the prostate tissue sections, and the authors hypothesized that the incomplete treatment of the prostate tissue could be due to an insufficient light power delivered to cancer cells. This conclusion emphasizes two important points: (a) development of a PDT modality which does not require the external illumination which is strongly attenuated by the overlaying tissue layers would be a significant advance, however (b) at the same time, sufficient power of the upconverted emission light is required for the PDT to be successful. A number of recent publications quote local light power of $1.5 \text{ mW}\cdot\text{cm}^{-2}$ to $2.5 \text{ mW}\cdot\text{cm}^{-2}$ as the lowest power required for PDT to work. Unfortunately, the manuscript does not provide the power of luminescent light achievable with the up-conversion nanoparticles.*

Response: The question the reviewer raised is very important. Indeed, the unsatisfactory treatment effect of PDT for prostate cancer is primarily attributable to tissue coverage dissipating the majority of energy. Previous studies have reported that 980 nm NIR energy significantly decreases as it passes through deep tissues, thereby reducing its tissue penetration depth. Additionally, the water in biological samples absorbs the wavelength spectrum in this region, which further diminishes its therapeutic effect^{3,4}. Regarding your concern about the effective power required for PDT, we have provided the power of luminescent light achievable with the UCNPs in the results section of

the manuscript (Supplementary Fig. 15). According to the results, NaGdF₄:Yb,Tm@CaF₂:Eu UCNPs requires a minimal power of approximately 0.3 W cm⁻² for PDT to work. In addition, as indicated in the manuscript, the irradiation power of STPNs is 2.0 W cm⁻² and our design avoids the energy attenuation caused by tissue obstruction. When STPNs accumulate in the tumor after 1 h, over 50% of the afterglow luminescence remains (Fig. 2j), and the ROS are still being generated by STPNs to damage the tumor cells (Fig. 2i). Therefore, the residual energy left in STPNs are theoretically sufficient to eliminate tumors. Furthermore, the animal model further validates the antitumor ability of STPNs (Fig. 5).

Supplementary Figure 15: a) Power-dependent UCL spectra of NaGdF₄:Yb,Tm@CaF₂:Eu core@shell UCNP under 980 nm laser irradiation b) UCL spectra of NaGdF₄:Yb,Tm@CaF₂:Eu core@shell UCNP under 980 nm laser irradiation at a power density of 0.3 W cm⁻².

Comments: *To overcome this problem and demonstrate that the approach works at least in the studies with the tumor-bearing mice, the authors indicate that tumor bioluminescence intensity in treated mice gradually decreased and disappeared. This is interpreted as an excellent tumor suppressive efficacy of the treatment. It is not entirely clear why bioluminescence intensity is used to demonstrate that PDT treatment works when an obvious gold standard would be the histopathological analysis of tissue of tumor-bearing and control mice after they are sacrificed.*

Response: Thank you for your pointing this out. First, we agree that histopathological analysis of tumor-bearing and control mice's sacrificed tissue should be the gold standard for measuring treatment efficacy. We demonstrated H&E and Ki67 staining results of orthotopic GBC tissues in different groups after treatments. Second, unlike subcutaneous tumors, in situ abdominal tumors cannot be observed directly. The lentivirus expressing the Luciferase gene is transmitted to tumor cells in advance, and under the catalysis of its substrate fluorescein sodium salt, tumor cells can spontaneously produce fluorescence, with the intensity of the fluorescence representing the number of tumor cells. Thus, we can use the IVIS Spectrum in vivo imaging system to observe the progression of tumors in mice during treatment. This technique has been used in several studies to monitor tumor growth and metastasis in vivo⁵⁻⁸.

Comments: *Moreover, it appears that mice were indeed sacrificed with H&E staining section analysis of vital organs showing no apparent differences between treated and untreated mice, posing obvious questions about the potential effect of the technique on tumor tissue.*

Response: We apologize for the confusion caused by the lack of distinct markings on the H&E staining and added red arrows to clarify this issue. In fact, H&E and Ki67 staining revealed distinct differences between treated and untreated mice.

Fig. 5e: Representative images of H&E staining of orthotopic GBC tissues in PBS or STPNs+NIR group (The red arrows show the obvious tumor sites; scale bar = 250 μ m).

Comments: *Also a minor point is that the GBC is indeed very deadly but relatively rare cancer and the authors might consider more common cancers involving the gastrointestinal tract, for example, pancreatic cancer.*

Response: Regarding the tumor model, pancreatic cancer is also a prevalent type of deep tumor. However, the five-year survival rate for pancreatic cancer has nearly doubled over the past decade due to a greater understanding of the disease's biological foundations^{9,10}. Current first-line systemic chemotherapy regimens for pancreatic cancer (including FOLFIRINOX and gemcitabine combined with paclitaxel) can significantly improve the five-year survival rate of patients. In contrast, GBC is the most prevalent malignant tumor of the biliary system that is typically fatal (5-year survival rate of less than 5%) due to the limited efficacy of existing treatments^{11,12,13,14}. Consequently, developing an effective targeted therapy for GBC remains essential and urgent. Our team has previously been working on targeted therapy for GBC. STPNs excellent antitumor activity in the GBC model in this study has significant implications for clinical practice. The extended application of STPNs in other deep tumor models was also explored in the discussion section of the manuscript. Additionally, the main reasons for using GBC as a model for deep-seated tumors are listed as follows. First, unlike tumors covered by superficial skin, such as breast cancer, GBC is a genuine deep tumor due to its location at the right upper abdomen, deep within the gallbladder fossa below the liver. Second, GBC is the most prevalent cancer of the biliary system with a poor prognosis^{11,12} and the sixth most widespread cancer of the gastrointestinal system. Although surgery is regarded as the most effective treatment for early GBC, nearly half of the patients suffer a postoperative recurrence within five years¹⁵⁻¹⁷. Most patients are usually diagnosed at advanced and later stages where surgical removal of the tumor may not be possible. Moreover, patients with advanced and recurrent GBC do not respond to conventional

chemotherapy and radiotherapy, resulting in a 5-year survival rate of less than 5%^{13,14}. Due to its highly malignant biology and challenging anatomic location, it is more challenging to access and treat with current therapies. Therefore, studying GBC can provide insights into the challenges of treating other deep-seated tumors.

Reviewer Comments to Author

Reviewer #3

Comments: *The limited depth of light penetration largely hinders the practical use of PDT in vivo. Sarun et al. present a pH responsive, size switchable nanoplatfrom (STPN) for PDT/trastuzumab synergistic therapy in orthotopic gallbladder cancer. The afterglow luminescence of STPN excited NIR-laser served as a persistent light source for PDT, which overcomes the problem of tissue penetration of laser. The photoactivity of photosensitizer in the STPNs was partially quenched and recovered in the acidic tumor environment, thus reducing the toxicity during in vivo circulation. This work is an important extension of previously published work by the Ling group (Adv. Mater. 2018, 30, 1802808). I would recommend its publication in Nature Communications, after the following concerns are addressed?*

Response: Thank you very much for your encouraging remarks. We have modified our manuscript based on your suggestions and cited the work published by Ling's group. We believe that the quality of our manuscript has been significantly improved by your comments.

Q1: *In Figure 2i, laser power should be given.*

Response: We have provided the laser power for Figure 2i in the figure legends and the methods section.

Q2: *Since the PDT is triggered by afterglow luminescence, the intensity and decay rate of afterglow luminescence is highly related with the final therapeutic outcome of PDT. The readers are more care about whether the afterglow luminescence is stronger or longer enough for effective PDT? It would be better to show more trials under various conditions (e.g. different laser power, different irradiation time)*

Response: The afterglow intensity of STPNs irradiated by different powers and duration of 980 nm laser source were evaluated in the discussion section of the manuscript (Supplementary Fig. 33).

Supplementary Figure 33: STPNs afterglow intensity at different irradiation time (2.5, 5, 10 min) under 980 nm laser irradiation at a power of a) 1.0 W cm⁻², b) 2.0 W cm⁻², and c) 4.0 W cm⁻².

Q3: It would be beneficial to discuss the corresponding time points between afterglow luminescence decay and in vivo tumor accumulation of STPNs.

Response: The corresponding time points between afterglow luminescence decay and in vivo tumor accumulation of STPNs have been discussed in the discussion section of the manuscript. As depicted in Supplementary Fig. 33, the afterglow intensity increases as irradiation power and duration increase. Interestingly, as the laser irradiation time increases at a power density of 4.0 W cm⁻², the afterglow intensity weakens. This is possibly due to the intense power that leads to the disintegration of the nanoparticle structure¹⁸. Therefore, based on these results and the accumulation of STPNs at the tumor site within 1 h, STPNs were exposed to a 980 nm laser source at a power density of 2.0 W cm⁻² for 5 min.

Q4: Semi-quantitative analysis of western-blot in Figure 4b should be given.

Response: A semi-quantitative analysis of the western blot in Fig. 4b has been provided.

Fig. 4b: Western blot for NFκB p65, NFκB p50, β-actin, and Lamin B1 in the cytoplasm or/and nucleus of GBC-SD or EH-GB1 cells exposed to normal media, TZB, SPNs, or STPNs with/without 980 nm laser irradiation (2.0 W cm⁻², 5 min with every 1 min interval).

References

- 1 Miao, Q. *et al.* Molecular afterglow imaging with bright, biodegradable polymer nanoparticles. *Nat Biotechnol* **35**, 1102-1110, doi:10.1038/nbt.3987 (2017).
- 2 Chen, W. *et al.* Near-Infrared Afterglow Luminescence of Chlorin Nanoparticles for Ultrasensitive In Vivo Imaging. *J Am Chem Soc* **144**, 6719-6726, doi:10.1021/jacs.1c10168 (2022).
- 3 Chen, G., Qiu, H., Prasad, P. N. & Chen, X. Upconversion nanoparticles: design, nanochemistry, and applications in theranostics. *Chem Rev* **114**, 5161-5214, doi:10.1021/cr400425h (2014).
- 4 Hang, Y., Boryczka, J. & Wu, N. Visible-light and near-infrared fluorescence and surface-enhanced Raman scattering point-of-care sensing and bio-imaging: a review. *Chem Soc Rev* **51**, 329-375, doi:10.1039/c9cs00621d (2022).
- 5 Zhang, D. *et al.* Near infrared-activatable biomimetic nanogels enabling deep tumor drug penetration inhibit orthotopic glioblastoma. *Nat Commun* **13**, 6835, doi:10.1038/s41467-022-34462-8 (2022).
- 6 Chen, Z. *et al.* PRDM8 exhibits antitumor activities toward hepatocellular carcinoma by targeting NAP1L1. *Hepatology (Baltimore, Md.)* **68**, 994-1009, doi:10.1002/hep.29890 (2018).
- 7 Jiang, B. *et al.* Lysosomal protein transmembrane 5 promotes lung-specific metastasis by regulating BMPR1A lysosomal degradation. *Nat Commun* **13**, 4141, doi:10.1038/s41467-022-31783-6 (2022).
- 8 Kanda, M. *et al.* SYT7 acts as a driver of hepatic metastasis formation of gastric cancer cells. *Oncogene* **37**, 5355-5366, doi:10.1038/s41388-018-0335-8 (2018).
- 9 Halbrook, C. J., Lyssiotis, C. A., Pasca di Magliano, M. & Maitra, A. Pancreatic cancer: Advances and challenges. *Cell* **186**, 1729-1754, doi:10.1016/j.cell.2023.02.014 (2023).
- 10 Jemal, A., Siegel, R., Xu, J. & Ward, E. Cancer statistics, 2010. *CA: a cancer journal for clinicians* **60**, 277-300, doi:10.3322/caac.20073 (2010).
- 11 Siegel, R. L., Miller, K. D., Wagle, N. S. & Jemal, A. Cancer statistics, 2023. *CA Cancer J Clin* **73**, 17-48, doi:10.3322/caac.21763 (2023).
- 12 Sung, H. *et al.* Global Cancer Statistics 2020: GLOBOCAN Estimates of Incidence and Mortality Worldwide for 36 Cancers in 185 Countries. *CA Cancer J Clin* **71**, 209-249, doi:10.3322/caac.21660 (2021).
- 13 Roa, J. C. *et al.* Gallbladder cancer. *Nat Rev Dis Primers* **8**, 69, doi:10.1038/s41572-022-00398-y (2022).
- 14 Hundal, R. & Shaffer, E. A. Gallbladder cancer: epidemiology and outcome. *Clin Epidemiol* **6**, 99-109, doi:10.2147/clep.S37357 (2014).
- 15 Shimizu, Y. *et al.* Early Recurrence in Resected Gallbladder Carcinoma: Clinical Impact and Its Preoperative Predictive Score. *Ann Surg Oncol* **29**, 5447-5457, doi:10.1245/s10434-022-11937-y (2022).
- 16 Benson, A. B. *et al.* Hepatobiliary Cancers, Version 2.2021, NCCN Clinical Practice Guidelines in Oncology. *J Natl Compr Canc Netw* **19**, 541-565, doi:10.6004/jnccn.2021.0022 (2021).
- 17 Chen, M. *et al.* Development and Validation of a Nomogram for Predicting Survival in Gallbladder Cancer Patients With Recurrence After Surgery. *Front Oncol* **10**, 537789, doi:10.3389/fonc.2020.537789 (2020).

- 18 Chen, M. *et al.* Bortezomib-Encapsulated Dual Responsive Copolymeric Nanoparticles for Gallbladder Cancer Targeted Therapy. *Adv Sci (Weinh)* **9**, e2103895, doi:10.1002/advs.202103895 (2022).

Reviewers' Comments:

Reviewer #1:

Remarks to the Author:

The revised paper can be accepted for publication.

Reviewer #2:

Remarks to the Author:

The authors have revised the manuscript, attempting to clarify the two main concerns posed by this reviewer in the previous review.

To answer the first concern that the manuscript does not provide the power of luminescent light achievable with the up-conversion nanoparticles, the authors now state on page 6 of the rebuttal letter that the irradiation power of STPNs is 2.0 W cm^{-2} . However, not only does this intensity appear to be unrealistically high but, if correct, it will cause significant thermal damage not only to the tumor cells but also to all the surrounding tissues in a significant volume adjacent to the nanoparticles. Taking into account that in the near infrared spectral region, the light transport length in tissue, which is almost entirely determined by the reduced scattering coefficient, is close to one millimeter, one can estimate that for an irradiation power of 2.0 W cm^{-2} , the temperature of the tissue will increase approximately by $10 \text{ }^\circ\text{C}$ every second. Moreover, as the characteristic time scale for heat diffusion is around 5 seconds for millimeter distances, the absorbed heat will not dissipate due to heat diffusion. Taking into account the normal body temperature for mammals, the $50 \text{ }^\circ\text{C}$ temperature increase will result in a temperature that is over $80 \text{ }^\circ\text{C}$ in a volume of several tens of cubic millimeters around the nanoparticles. This temperature will lead to full-thickness tissue destruction in seconds. This scenario is rather unrealistic and has not been observed in experiments. Therefore, the power of luminescent light achievable with the up-conversion nanoparticles appears to be significantly lower than that quoted in the response and is still unclear.

Regarding the second concern that it is not entirely clear why bioluminescence intensity is used to demonstrate that PDT treatment works while the gold standard is the histopathological analysis of tissue of tumor-bearing and control mice after they are sacrificed, the authors state that "abdominal tumors cannot be observed directly". This statement does not seem to be correct as histopathological analysis of tissue after animals are sacrificed is the standard evaluation approach for animal studies, such as the studies described in the manuscript.

Therefore the opinion of this reviewer remains that the manuscript presents intriguing but early and not particularly complete results. The answers provided by the authors in response to the review are unfortunately not very convincing. Therefore, the work is not of the far-reaching scope or of ground-breaking value typical of manuscripts published in the Nature family journals and this manuscript is better suited for a specialized journal.

Reviewer #3:

Remarks to the Author:

The authors have addressed all my comments and revised the manuscript accordingly. I'm happy to recommend its publication as is.

Reviewer Comments to Author

Reviewer #2

Comments: *The authors have revised the manuscript, attempting to clarify the two main concerns posed by this reviewer in the previous review.*

To answer the first concern that the manuscript does not provide the power of luminescent light achievable with the up-conversion nanoparticles, the authors now state on page 6 of the rebuttal letter that the irradiation power of STPNs is 2.0 W cm^{-2} . However, not only does this intensity appear to be unrealistically high but, if correct, it will cause significant thermal damage not only to the tumor cells but also to all the surrounding tissues in a significant volume adjacent to the nanoparticles. Taking into account that in the near infrared spectral region, the light transport length in tissue, which is almost entirely determined by the reduced scattering coefficient, is close to one millimeter, one can estimate that for an irradiation power of 2.0 W cm^{-2} , the temperature of the tissue will increase approximately by $10 \text{ }^\circ\text{C}$ every second. Moreover, as the characteristic time scale for heat diffusion is around 5 seconds for millimeter distances, the absorbed heat will not dissipate due to heat diffusion. Taking into account the normal body temperature for mammals, the $50 \text{ }^\circ\text{C}$ temperature increase will result in a temperature that is over $80 \text{ }^\circ\text{C}$ in a volume of several tens of cubic millimeters around the nanoparticles. This temperature will lead to full-thickness tissue destruction in seconds. This scenario is rather unrealistic and has not been observed in experiments. Therefore, the power of luminescent light achievable with the up-conversion nanoparticles appears to be significantly lower than that quoted in the response and is still unclear.

Response: We apologize for the confusion due to the unclear description. In this study, STPNs were only pre-irradiated with a 980 nm laser ex vivo at a power density of 2.0 W cm^{-2} for 5 min before treatment. Although the temperature increases to $\sim 75^\circ\text{C}$ during laser irradiation, the temperature rapidly drops to body temperature ($\sim 37^\circ\text{C}$) 2-3 min after laser irradiation stops. Generally, the temperature will cool down to a safe level over time. Afterward, STPNs were injected into the mice and the subsequent emission of Pu18 does not generate heat (Fig. 1). Theoretically, damage to the surrounding tissues is not likely to occur.

In addition, we have previously evaluated the afterglow intensity of STPNs at different irradiation times and laser power in our study and found the optimal laser duration and power (2.0 W cm^{-2} , 5 min). As depicted in Supplementary Fig. 33, the afterglow intensity is significantly weaker at a low-power density of 1.0 W cm^{-2} , whereas the afterglow intensity weakens as the laser irradiation time increases at a high-power density of 4.0 W cm^{-2} due to the intense power that leads to the disintegration of the nanoparticle structure¹⁻⁴. Therefore, based on these results and the accumulation of STPNs at the tumor site within 1 h, STPNs were exposed to a 980 nm laser ex vivo at a power density of 2.0 W cm^{-2} for 5 min before treatment.

Overall, as emphasized in the manuscript, the unique design of ex vivo pre-activated STPNs before treatment can bypass the impenetrable tissue thickness of NIR light and avoid the heating of the surrounding tissues induced by 980 nm laser irradiation.

Figure 1: Schematic illustration of STPNs treatment procedure under 980 nm laser irradiation at a power density of 2.0 W cm^{-2} for 5 min and its corresponding temperature changes.

Supplementary Figure 33: STPNs afterglow intensity at different irradiation time (2.5, 5, 10 min) under 980 nm laser irradiation at a power of a) 1.0 W cm^{-2} , b) 2.0 W cm^{-2} , and c) 4.0 W cm^{-2} .

Comments: *Regarding the second concern that it is not entirely clear why bioluminescence intensity is used to demonstrate that PDT treatment works while the gold standard is the histopathological analysis of tissue of tumor-bearing and control mice after they are sacrificed, the authors state that “abdominal tumors cannot be observed directly”. This statement does not seem to be correct as histopathological analysis of tissue after animals are sacrificed is the standard evaluation approach for animal studies, such as the studies described in the manuscript.*

Response: We agree that the gold standard for treatment efficacy evaluation is histopathological analysis. In fact, we evaluated the treatment efficacy based on the histopathological analyses after treatment and bioluminescence imaging during treatment (Fig. 2). These two experiments have well demonstrated the anti-tumor effect of STPNs (Fig. 5B and E).

Figure 2: Schematic illustration of treatment procedure.

Fig. 5e: Representative images of H&E and Ki67 staining of orthotopic GBC tissues in different groups after treatments (scale bar = 250 μ m).

Fig. 5b: In vivo bioluminescence (BL) images of orthotopic GBC in live mice receiving PBS, TZB, STPNs, or STPNs irradiated by 980 nm laser source before injection at a power density of 2.0 W cm^{-2} for 5 min.

References

- 1 Chen, W. *et al.* Near-Infrared Afterglow Luminescence of Chlorin Nanoparticles for Ultrasensitive In Vivo Imaging. *J Am Chem Soc* **144**, 6719-6726, doi:10.1021/jacs.1c10168 (2022).
- 2 Chen, M. *et al.* Bortezomib-Encapsulated Dual Responsive Copolymeric Nanoparticles for Gallbladder Cancer Targeted Therapy. *Adv Sci (Weinh)* **9**, e2103895, doi:10.1002/advs.202103895 (2022).
- 3 Li, X. *et al.* Hypoxia-reinforced antitumor RNA interference mediated by micelleplexes with programmed disintegration. *Acta Biomater* **148**, 194-205, doi:https://doi.org/10.1016/j.actbio.2022.05.050 (2022).
- 4 Yan, B., Boyer, J.-C., Branda, N. R. & Zhao, Y. Near-Infrared Light-Triggered Dissociation of Block Copolymer Micelles Using Upconverting Nanoparticles. *Journal of the American Chemical Society* **133**, 19714-19717, doi:10.1021/ja209793b (2011).

Reviewers' Comments:

Reviewer #2:

Remarks to the Author:

My concerns have now been adequately addressed.